# *This* Looks Like *That*: Deep Learning for Interpretable Image Recognition

**Chaofan Chen**[*]
Duke University
cfchen@cs.duke.edu

**Oscar Li**[*]
Duke University
oscarli@alumni.duke.edu

**Chaofan Tao**
Duke University
chaofan.tao@duke.edu

**Alina Jade Barnett**
Duke University
abarnett@cs.duke.edu

**Jonathan Su**
MIT Lincoln Laboratory[†]
su@ll.mit.edu

**Cynthia Rudin**
Duke University
cynthia@cs.duke.edu

## Abstract

When we are faced with challenging image classification tasks, we often explain our reasoning by dissecting the image, and pointing out prototypical aspects of one class or another. The mounting evidence for each of the classes helps us make our final decision. In this work, we introduce a deep network architecture – *prototypical part network* (ProtoPNet), that reasons in a similar way: the network dissects the image by finding prototypical parts, and combines evidence from the prototypes to make a final classification. The model thus reasons in a way that is qualitatively similar to the way ornithologists, physicians, and others would explain to people on how to solve challenging image classification tasks. The network uses only image-level labels for training without any annotations for parts of images. We demonstrate our method on the CUB-200-2011 dataset and the Stanford Cars dataset. Our experiments show that ProtoPNet can achieve comparable accuracy with its analogous non-interpretable counterpart, and when several ProtoPNets are combined into a larger network, it can achieve an accuracy that is on par with some of the best-performing deep models. Moreover, ProtoPNet provides a level of interpretability that is absent in other interpretable deep models.

## 1 Introduction

How would you describe why the image in Figure 1 looks like a clay colored sparrow? Perhaps the bird's head and wing bars look like those of a prototypical clay colored sparrow. When we describe how we classify images, we might focus on parts of the image and compare them with prototypical parts of images from a given class. This method of reasoning is commonly used in difficult identification tasks: e.g., radiologists compare suspected tumors in X-ray scans with prototypical tumor images for diagnosis of cancer [13]. The question is whether we can ask a machine learning model to imitate this way of thinking, and to explain its reasoning process in a human-understandable way.

The goal of this work is to define a form of interpretability in image processing (*this* looks like *that*) that agrees with the way humans describe their own thinking in classification tasks. In this work,

---

[*]Contributed equally

[†]DISTRIBUTION STATEMENT A. Approved for public release. Distribution is unlimited. This material is based upon work supported by the Under Secretary of Defense for Research and Engineering under Air Force Contract No. FA8702-15-D-0001. Any opinions, findings, conclusions or recommendations expressed in this material are those of the author(s) and do not necessarily reflect the views of the Under Secretary of Defense for Research and Engineering.

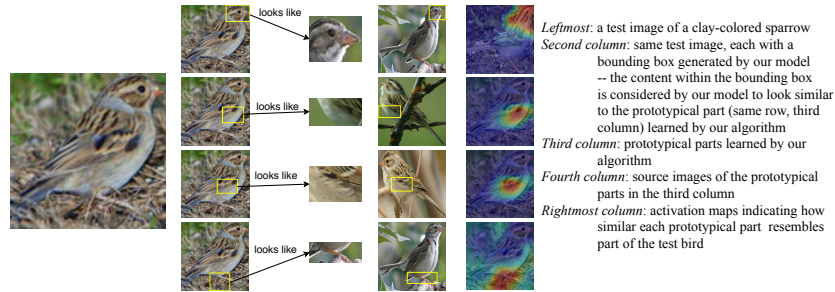

*Leftmost*: a test image of a clay-colored sparrow
*Second column*: same test image, each with a bounding box generated by our model -- the content within the bounding box is considered by our model to look similar to the prototypical part (same row, third column) learned by our algorithm
*Third column*: prototypical parts learned by our algorithm
*Fourth column*: source images of the prototypical parts in the third column
*Rightmost column*: activation maps indicating how similar each prototypical part resembles part of the test bird

Figure 1: Image of a clay colored sparrow and how parts of it look like some learned prototypical parts of a clay colored sparrow used to classify the bird's species.

we introduce a network architecture – *prototypical part network* (ProtoPNet), that accommodates this definition of interpretability, where comparison of image parts to learned prototypes is integral to the way our network reasons about new examples. Given a new bird image as in Figure 1, our model is able to identify several parts of the image where it thinks that *this* part of the image looks like *that* prototypical part of some class, and makes its prediction based on a weighted combination of the similarity scores between parts of the image and the learned prototypes. In this way, our model is *interpretable*, in the sense that it has a transparent reasoning process when making predictions. Our experiments show that our ProtoPNet can achieve comparable accuracy with its analogous non-interpretable counterpart, and when several ProtoPNets are combined into a larger network, our model can achieve an accuracy that is on par with some of the best-performing deep models. Moreover, our ProtoPNet provides a level of interpretability that is absent in other interpretable deep models.

Our work relates to (but contrasts with) those that perform *posthoc* interpretability analysis for a trained convolutional neural network (CNN). In posthoc analysis, one interprets a trained CNN by fitting explanations to how it performs classification. Examples of posthoc analysis techniques include activation maximization [5, 12, 22, 44, 30, 38, 50], deconvolution [51], and saliency visualization [38, 42, 41, 36]. All of these posthoc visualization methods do not explain the reasoning process of how a network *actually* makes its decisions. In contrast, our network has a built-in case-based reasoning process, and the explanations generated by our network are actually used during classification and are not created posthoc.

Our work relates closely to works that build attention-based interpretability into CNNs. These models aim to expose the parts of an input the network focuses on when making decisions. Examples of attention models include class activation maps [56] and various part-based models (e.g., [55, 53, 15, 57, 43, 10, 9, 34, 37, 49, 7]; see Table 1). However, attention-based models can only tell us *which parts* of the input they are looking at – they do not point us to prototypical cases to which the parts they focus on are similar. On the other hand, our ProtoPNet is not only able to expose the parts of the input it is looking at, but also point us to prototypical cases similar to those parts. Section 2.5 provides a comparison between attention-based models and our ProtoPNet.

Recently there have also been attempts to quantify the interpretability of visual representations in a CNN, by measuring the overlap between highly activated image regions and labeled visual concepts [1, 54]. However, to quantitatively measure the interpretability of a convolutional unit in a network requires fine-grained labeling for a significantly large dataset specific to the purpose of the network. The existing Broden dataset for scene/object classification networks [1] is not well-suited to measure the unit interpretability of a network trained for fine-grained classification (which is our main application), because the concepts detected by that network may not be present in the Broden dataset. Hence, in our work, we do not focus on quantifying unit interpretability of our network, but instead look at the *reasoning process* of our network which is qualitatively similar to that of humans.

Our work uses generalized convolution [8, 29] by including a prototype layer that computes squared $L^2$ distance instead of conventional inner product. In addition, we propose to constrain each convolutional filter to be *identical* to some latent training patch. This added constraint allows us to interpret the convolutional filters as visualizable prototypical image parts and also necessitates a novel training procedure.

Our work relates closely to other case-based classification techniques using k-nearest neighbors [47, 35, 32] or prototypes [33, 2, 48], and very closely, to the Bayesian Case Model [18]. It relates to traditional "bag-of-visual-words" models used in image recognition [21, 6, 17, 40, 31]. These models (like our ProtoPNet) also learn a set of prototypical parts for comparison with an unseen image. However, the feature extraction in these models is performed by Scale Invariant Feature Transform (SIFT) [27], and the learning of prototypical patches ("visual words") is done separately from the feature extraction (and the learning of the final classifier). In contrast, our ProtoPNet uses a specialized neural network architecture for feature extraction and prototype learning, and can be trained in an *end-to-end* fashion. Our work also relates to works (e.g., [3, 24]) that identify a set of prototypes for pose alignment. However, their prototypes are templates for warping images and similarity with these prototypes does not provide an explanation for why an image is classified in a certain way. Our work relates most closely to Li et al. [23], who proposed a network architecture that builds case-based reasoning into a neural network. However, their model requires a decoder (for visualizing prototypes), which fails to produce realistic prototype images when trained on datasets of natural images. In contrast, our model does not require a decoder for prototype visualization. Every prototype is the latent representation of some training image patch, which naturally and faithfully becomes the prototype's visualization. The removal of the decoder also facilitates the training of our network, leading to better explanations and better accuracy. Unlike the work of Li et al., whose prototypes represent entire images, our model's prototypes can have much smaller spatial dimensions and represent *prototypical parts of images*. This allows for more fine-grained comparisons because different parts of an image can now be compared to different prototypes. Ming et al. [28] recently took the concepts in [23] and the preprint of an earlier version of this work, which both involve integrating prototype learning into CNNs for image recognition, and used these concepts to develop prototype learning in recurrent neural networks for modeling sequential data.

## 2 Case study 1: bird species identification

In this case study, we introduce the architecture and the training procedure of our ProtoPNet in the context of bird species identification, and provide a detailed walk-through of how our network classifies a new bird image and explains its prediction. We trained and evaluated our network on the CUB-200-2011 dataset [45] of 200 bird species. We performed offline data augmentation, and trained on images cropped using the bounding boxes provided with the dataset.

### 2.1 ProtoPNet architecture

Figure 2 gives an overview of the architecture of our ProtoPNet. Our network consists of a regular convolutional neural network $f$, whose parameters are collectively denoted by $w_{\text{conv}}$, followed by a prototype layer $g_{\mathbf{p}}$ and a fully connected layer $h$ with weight matrix $w_h$ and no bias. For the regular convolutional network $f$, our model use the convolutional layers from models such as VGG-16, VGG-19 [39], ResNet-34, ResNet-152 [11], DenseNet-121, or DenseNet-161 [14] (initialized with filters pretrained on ImageNet [4]), followed by two additional $1 \times 1$ convolutional layers in our experiments. We use ReLU as the activation function for all convolutional layers except the last for which we use the sigmoid activation function.

Given an input image $\mathbf{x}$ (such as the clay colored sparrow in Figure 2), the convolutional layers of our model extract useful features $f(\mathbf{x})$ to use for prediction. Let $H \times W \times D$ be the shape of the convolutional output $f(\mathbf{x})$. For the bird dataset with input images resized to $224 \times 224 \times 3$, the spatial dimension of the convolutional output is $H = W = 7$, and the number of output channels $D$ in the additional convolutional layers is chosen from three possible values: 128, 256, 512, using cross validation. The network learns $m$ prototypes $\mathbf{P} = \{\mathbf{p}_j\}_{j=1}^m$, whose shape is $H_1 \times W_1 \times D$ with $H_1 \leq H$ and $W_1 \leq W$. In our experiments, we used $H_1 = W_1 = 1$. Since the depth of each prototype is the same as that of the convolutional output but the height and the width of each prototype is smaller than those of the whole convolutional output, each prototype will be used to represent some prototypical activation pattern in a *patch* of the convolutional output, which in turn will correspond to some prototypical image patch in the original pixel space. Hence, each prototype $\mathbf{p}_j$ can be understood as the latent representation of some prototypical *part* of some bird image in this case study. As a schematic illustration, the first prototype $\mathbf{p}_1$ in Figure 2 corresponds to the head of a clay colored sparrow, and the second prototype $\mathbf{p}_2$ the head of a Brewer's sparrow. Given a convolutional output $\mathbf{z} = f(\mathbf{x})$, the $j$-th prototype unit $g_{\mathbf{p}_j}$ in the prototype layer $g_{\mathbf{p}}$ computes the

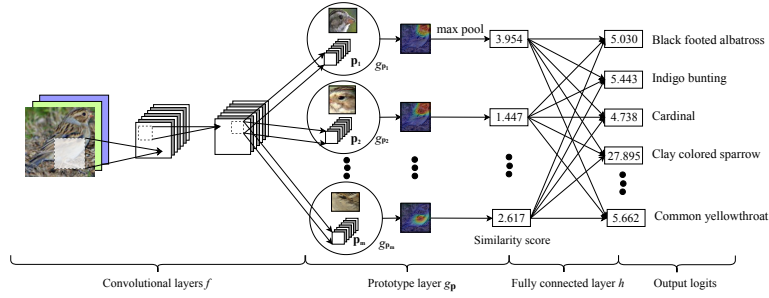

Figure 2: ProtoPNet architecture.

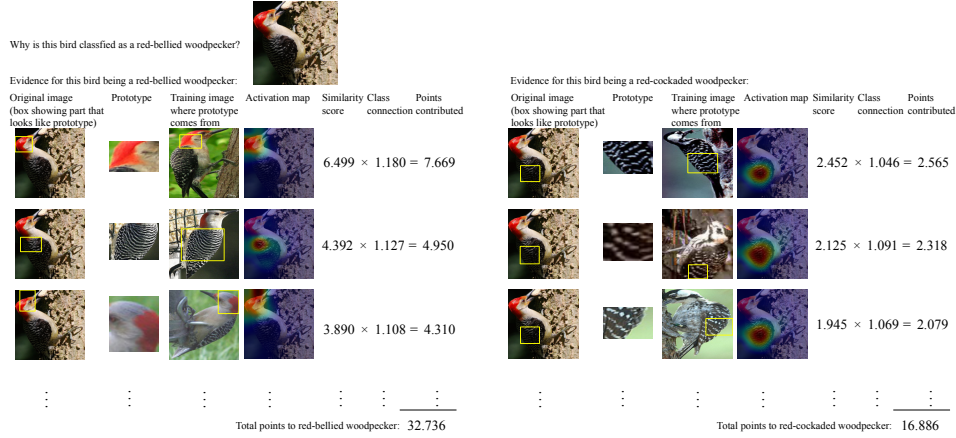

Figure 3: The reasoning process of our network in deciding the species of a bird (top).

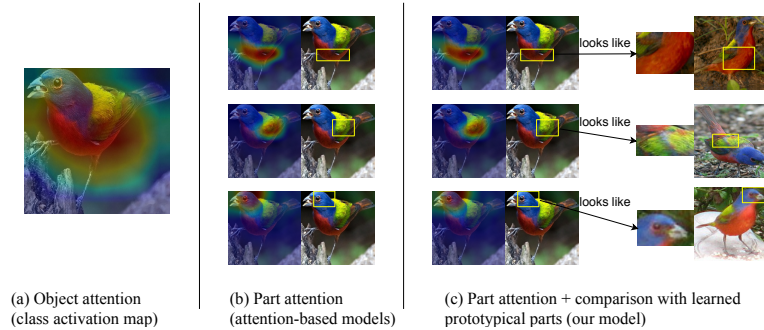

(a) Object attention
(class activation map)

(b) Part attention
(attention-based models)

(c) Part attention + comparison with learned
prototypical parts (our model)

Figure 4: Visual comparison of different types of model interpretability: (a) object-level attention map (e.g., class activation map [56]); (b) part attention (provided by attention-based interpretable models); and (c) part attention with similar prototypical parts (provided by our model).

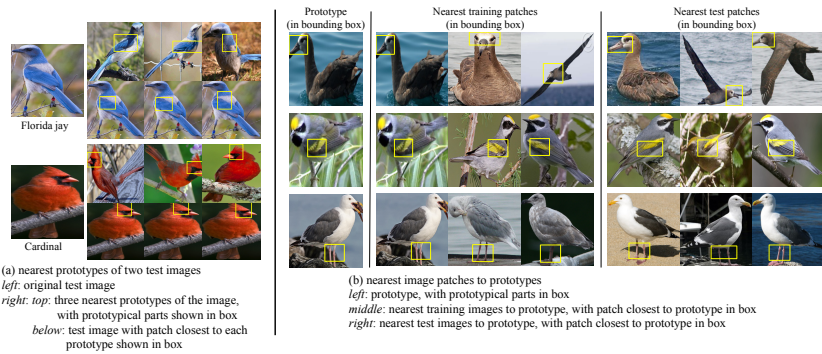

(a) nearest prototypes of two test images
*left*: original test image
*right*: *top*: three nearest prototypes of the image,
        with prototypical parts shown in box
      *below*: test image with patch closest to each
        prototype shown in box

(b) nearest image patches to prototypes
*left*: prototype, with prototypical parts in box
*middle*: nearest training images to prototype, with patch closest to prototype in box
*right*: nearest test images to prototype, with patch closest to prototype in box

Figure 5: Nearest prototypes to images and nearest images to prototypes. The prototypes are learned from the training set.

squared $L^2$ distances between the $j$-th prototype $\mathbf{p}_j$ and all patches of $\mathbf{z}$ that have the same shape as $\mathbf{p}_j$, and inverts the distances into similarity scores. The result is an activation map of similarity scores whose value indicates how strong a prototypical part is present in the image. This activation map preserves the spatial relation of the convolutional output, and can be upsampled to the size of the input image to produce a heat map that identifies which part of the input image is most similar to the learned prototype. The activation map of similarity scores produced by each prototype unit $g_{\mathbf{p}_j}$ is then reduced using global max pooling to a single similarity score, which can be understood as how strongly a prototypical part is present in *some* patch of the input image. In Figure 2, the similarity score between the first prototype $\mathbf{p}_1$, a clay colored sparrow head prototype, and the most activated (upper-right) patch of the input image of a clay colored sparrow is $3.954$, and the similarity score between the second prototype $\mathbf{p}_2$, a Brewer's sparrow head prototype, and the most activated patch of the input image is $1.447$. This shows that our model finds that the head of a clay colored sparrow has a stronger presence than that of a Brewer's sparrow in the input image. Mathematically, the prototype unit $g_{\mathbf{p}_j}$ computes $g_{\mathbf{p}_j}(\mathbf{z}) = \max_{\tilde{\mathbf{z}} \in \text{patches}(\mathbf{z})} \log \left( (\|\tilde{\mathbf{z}} - \mathbf{p}_j\|_2^2 + 1)/(\|\tilde{\mathbf{z}} - \mathbf{p}_j\|_2^2 + \epsilon) \right)$. The function $g_{\mathbf{p}_j}$ is *monotonically decreasing* with respect to $\|\tilde{\mathbf{z}} - \mathbf{p}_j\|_2$ (if $\tilde{\mathbf{z}}$ is the closest latent patch to $\mathbf{p}_j$). Hence, if the output of the $j$-th prototype unit $g_{\mathbf{p}_j}$ is large, then there is a patch in the convolutional output that is (in 2-norm) very close to the $j$-th prototype in the latent space, and this in turn means that there is a patch in the input image that has a similar concept to what the $j$-th prototype represents.

In our ProtoPNet, we allocate a pre-determined number of prototypes $m_k$ for each class $k \in \{1, ..., K\}$ (10 per class in our experiments), so that every class will be represented by some prototypes in the final model. Section S9.2 of the supplement discusses the choice of $m_k$ and other hyperparameters in greater detail. Let $\mathbf{P}_k \subseteq \mathbf{P}$ be the subset of prototypes that are allocated to class $k$: these prototypes should capture the most relevant parts for identifying images of class $k$.

Finally, the $m$ similarity scores produced by the prototype layer $g_{\mathbf{p}}$ are multiplied by the weight matrix $w_h$ in the fully connected layer $h$ to produce the output logits, which are normalized using softmax to yield the predicted probabilities for a given image belonging to various classes.

ProtoPNet's inference computation mechanism can be viewed as a special case of a more general type of probabilistic inference under some reasonable assumptions. This interpretation is presented in detail in Section S2 of the supplementary material.

## 2.2 Training algorithm

The training of our ProtoPNet is divided into: (1) stochastic gradient descent (SGD) of layers before the last layer; (2) projection of prototypes; (3) convex optimization of last layer. It is possible to cycle through these three stages more than once. The entire training algorithm is summarized in an algorithm chart, which can be found in Section S9.3 of the supplement.

**Stochastic gradient descent (SGD) of layers before last layer:** In the first training stage, we aim to learn a meaningful latent space, where the most important patches for classifying images are clustered (in $L^2$-distance) around semantically similar prototypes of the images' true classes, and the clusters that are centered at prototypes from different classes are well-separated. To achieve this goal, we jointly optimize the convolutional layers' parameters $w_{\text{conv}}$ and the prototypes $\mathbf{P} = \{\mathbf{p}_j\}_{j=1}^m$ in the prototype layer $g_{\mathbf{p}}$ using SGD, while keeping the last layer weight matrix $w_h$ fixed. Let $D = [\mathbf{X}, \mathbf{Y}] = \{(\mathbf{x}_i, y_i)\}_{i=1}^n$ be the set of training images. The optimization problem we aim to solve here is:

$$\min_{\mathbf{P}, w_{\text{conv}}} \frac{1}{n} \sum_{i=1}^n \text{CrsEnt}(h \circ g_{\mathbf{p}} \circ f(\mathbf{x_i}), \mathbf{y_i}) + \lambda_1 \text{Clst} + \lambda_2 \text{Sep}, \quad \text{where Clst and Sep are defined by}$$

$$\text{Clst} = \frac{1}{n} \sum_{i=1}^n \min_{j:\mathbf{p}_j \in \mathbf{P}_{y_i}} \min_{\mathbf{z} \in \text{patches}(f(\mathbf{x}_i))} \|\mathbf{z} - \mathbf{p}_j\|_2^2; \text{Sep} = -\frac{1}{n} \sum_{i=1}^n \min_{j:\mathbf{p}_j \notin \mathbf{P}_{y_i}} \min_{\mathbf{z} \in \text{patches}(f(\mathbf{x}_i))} \|\mathbf{z} - \mathbf{p}_j\|_2^2.$$

The cross entropy loss (CrsEnt) penalizes misclassification on the training data. The minimization of the cluster cost (Clst) encourages each training image to have some latent patch that is close to at least one prototype of its own class, while the minimization of the separation cost (Sep) encourages every latent patch of a training image to stay away from the prototypes *not* of its own class. These terms shape the latent space into a semantically meaningful clustering structure, which facilitates the $L^2$-distance-based classification of our network.

In this training stage, we also fix the last layer $h$, whose weight matrix is $w_h$. Let $w_h^{(k,j)}$ be the $(k,j)$-th entry in $w_h$ that corresponds to the weight connection between the output of the $j$-th prototype unit $g_{\mathbf{p}_j}$ and the logit of class $k$. Given a class $k$, we set $w_h^{(k,j)} = 1$ for all $j$ with $\mathbf{p}_j \in \mathbf{P}_k$ and $w_h^{(k,j)} = -0.5$ for all $j$ with $\mathbf{p}_j \notin \mathbf{P}_k$ (when we are in this stage for the first time). Intuitively, the positive connection between a class $k$ prototype and the class $k$ logit means that similarity to a class $k$ prototype should increase the predicted probability that the image belongs to class $k$, and the negative connection between a non-class $k$ prototype and the class $k$ logit means that similarity to a non-class $k$ prototype should decrease class $k$'s predicted probability. By fixing the last layer $h$ in this way, we can force the network to learn a meaningful latent space because if a latent patch of a class $k$ image is too close to a non-class $k$ prototype, it will decrease the predicted probability that the image belongs to class $k$ and increase the cross entropy loss in the training objective. Note that both the separation cost and the negative connection between a non-class $k$ prototype and the class $k$ logit encourage prototypes of class $k$ to represent semantic concepts that are characteristic of class $k$ but not of other classes: if a class $k$ prototype represents a semantic concept that is also present in a non-class $k$ image, this non-class $k$ image will highly activate that class $k$ prototype, and this will be penalized by increased (i.e., less negative) separation cost and increased cross entropy (as a result of the negative connection). The separation cost is new to this paper, and has not been explored by previous works of prototype learning (e.g., [3, 23]).

**Projection of prototypes:** To be able to visualize the prototypes as training image patches, we project ("push") each prototype $\mathbf{p}_j$ onto the nearest latent training patch from the *same* class as that of $\mathbf{p}_j$. In this way, we can conceptually equate each prototype with a training image patch. (Section 2.3 discusses how we visualize the projected prototypes.) Mathematically, for prototype $\mathbf{p}_j$ of class $k$, i.e., $\mathbf{p}_j \in \mathbf{P}_k$, we perform the following update:

$$\mathbf{p}_j \leftarrow \arg\min_{\mathbf{z} \in \mathcal{Z}_j} \|\mathbf{z} - \mathbf{p}_j\|_2, \text{ where } \mathcal{Z}_j = \{\tilde{\mathbf{z}} : \tilde{\mathbf{z}} \in \text{patches}(f(\mathbf{x}_i)) \ \forall i \text{ s.t. } y_i = k\}.$$

The following theorem provides some theoretical understanding of how prototype projection affects classification accuracy. We use another notation for prototypes $\mathbf{p}_l^k$, where $k$ represents the class identity of the prototype and $l$ is the index of that prototype among all prototypes of that class.

**Theorem 2.1.** *Let $h \circ g_{\mathbf{p}} \circ f$ be a ProtoPNet. For each $k$, $l$, we use $\mathbf{b}_l^k$ to denote the value of the $l$-th prototype for class $k$ **before** the projection of $\mathbf{p}_l^k$ to the nearest latent training patch of class $k$, and use $\mathbf{a}_l^k$ to denote its value **after** the projection. Let $\mathbf{x}$ be an input image that is correctly classified by the ProtoPNet before the projection, $\mathbf{z}_l^k = \arg\min_{\tilde{\mathbf{z}} \in \text{patches}(f(\mathbf{x}))} \|\tilde{\mathbf{z}} - \mathbf{b}_l^k\|_2$ be the nearest patch of $f(\mathbf{x})$ to the prototype $\mathbf{p}_l^k$ before the projection (i.e., $\mathbf{b}_l^k$), and $c$ be the correct class label of $\mathbf{x}$.*

*Suppose that: (A1) $\mathbf{z}_l^k$ is also the nearest latent patch to prototype $\mathbf{p}_l^k$ after the projection ($\mathbf{a}_l^k$), i.e., $\mathbf{z}_l^k = \arg\min_{\tilde{\mathbf{z}} \in \text{patches}(f(\mathbf{x}))} \|\tilde{\mathbf{z}} - \mathbf{a}_l^k\|_2$; (A2) there exists some $\delta$ with $0 < \delta < 1$ such that: (A2a) for all incorrect classes' prototypes $k \neq c$ and $l \in \{1, ..., m_k\}$, we have $\|\mathbf{a}_l^k - \mathbf{b}_l^k\|_2 \leq \theta\|\mathbf{z}_l^k - \mathbf{b}_l^k\|_2 - \sqrt{\epsilon}$, where we define $\theta = \min\left(\sqrt{1+\delta} - 1, 1 - \frac{1}{\sqrt{2-\delta}}\right)$ ($\epsilon$ comes from the prototype activation function $g_{\mathbf{p}_j}$ defined in Section 2.1); (A2b) for the correct class $c$ and for all $l \in \{1, ..., m_c\}$, we have $\|\mathbf{a}_l^c - \mathbf{b}_l^c\|_2 \leq (\sqrt{1+\delta} - 1)\|\mathbf{z}_l^c - \mathbf{b}_l^c\|_2$ and $\|\mathbf{z}_l^c - \mathbf{b}_l^c\|_2 \leq \sqrt{1-\delta}$; (A3) the number of prototypes is the same for each class, which we denote by $m'$. (A4) for each class $k$, the weight connection in the fully connected last layer $h$ between a class $k$ prototype and the class $k$ logit is 1, and that between a non-class $k$ prototype and the class $k$ logit is 0 (i.e., $w_h^{(k,j)} = 1$ for all $j$ with $\mathbf{p}_j \in \mathbf{P}_k$ and $w_h^{(k,j)} = 0$ for all $j$ with $\mathbf{p}_j \notin \mathbf{P}_k$).*

*Then after projection, the output logit for the correct class $c$ can decrease at most by $\Delta_{\max} = m' \log((1+\delta)(2-\delta))$, and the output logit for every incorrect class $k \neq c$ can increase at most by $\Delta_{\max}$. If the output logits between the top-2 classes are at least $2\Delta_{\max}$ apart, then the projection of prototypes to their nearest latent training patches does not change the prediction of $\mathbf{x}$.*

Intuitively speaking, the theorem states that, if prototype projection does not move the prototypes by much (assured by the optimization of the cluster cost Clst), the prediction does not change for examples that the model predicted correctly with some confidence before the projection. The proof is in Section S1 of the supplement.

Note that prototype projection has the same time complexity as feedforward computation of a regular convolutional layer followed by global average pooling, a configuration common in standard CNNs

(e.g., ResNet, DenseNet), because the former takes the minimum distance over all prototype-sized patches, and the latter takes the average of dot-products over all filter-sized patches. Hence, prototype projection does not introduce extra time complexity in training our network.

**Convex optimization of last layer:** In this training stage, we perform a convex optimization on the weight matrix $w_h$ of last layer $h$. The goal of this stage is to adjust the last layer connection $w_h^{(k,j)}$, so that for $k$ and $j$ with $\mathbf{p}_j \notin \mathbf{P}_k$, our final model has the sparsity property $w_h^{(k,j)} \approx 0$ (initially fixed at $-0.5$). This sparsity is desirable because it means that our model relies less on a *negative* reasoning process of the form "this bird is of class $k'$ because it is *not* of class $k$ (it contains a patch that is *not* prototypical of class $k$)." The optimization problem we solve here is: $\min_{w_h} \frac{1}{n} \sum_{i=1}^{n} \text{CrsEnt}(h \circ g_{\mathbf{p}} \circ f(\mathbf{x_i}), \mathbf{y_i}) + \lambda \sum_{k=1}^{K} \sum_{j:\mathbf{p}_j \notin \mathbf{P}_k} |w_h^{(k,j)}|$. This optimization is convex because we fix all the parameters from the convolutional and prototype layers. This stage further improves accuracy without changing the learned latent space or prototypes.

## 2.3 Prototype visualization

Given a prototype $\mathbf{p}_j$ and the training image $\mathbf{x}$ whose latent patch is used as $\mathbf{p}_j$ during prototype projection, how do we decide which patch of $\mathbf{x}$ (in the pixel space) corresponds to $\mathbf{p}_j$? In our work, we use the image patch of $\mathbf{x}$ that is highly activated by $\mathbf{p}_j$ as the visualization of $\mathbf{p}_j$. The reason is that the patch of $\mathbf{x}$ that corresponds to $\mathbf{p}_j$ should be the one that $\mathbf{p}_j$ activates most strongly on, and we can find the patch of $\mathbf{x}$ on which $\mathbf{p}_j$ has the strongest activation by forwarding $\mathbf{x}$ through a trained ProtoPNet and upsampling the activation map produced by the prototype unit $g_{\mathbf{p}_j}$ (before max-pooling) to the size of the image $\mathbf{x}$ – the most activated patch of $\mathbf{x}$ is indicated by the high activation region in the (upsampled) activation map. We then visualize $\mathbf{p}_j$ with the smallest rectangular patch of $\mathbf{x}$ that encloses pixels whose corresponding activation value in the upsampled activation map from $g_{\mathbf{p}_j}$ is at least as large as the 95th-percentile of all activation values in that same map. Section S7 of the supplement describes prototype visualization in greater detail.

## 2.4 Reasoning process of our network

Figure 3 shows the reasoning process of our ProtoPNet in reaching a classification decision on a test image of a red-bellied woodpecker at the top of the figure. Given this test image $\mathbf{x}$, our model compares its latent features $f(\mathbf{x})$ against the learned prototypes. In particular, for each class $k$, our network tries to find evidence for $x$ to be of class $k$ by comparing its latent patch representations with every learned prototype $\mathbf{p}_j$ of class $k$. For example, in Figure 3 (left), our network tries to find evidence for the red-bellied woodpecker class by comparing the image's latent patches with each prototype (visualized in "Prototype" column) of that class. This comparison produces a map of similarity scores towards each prototype, which was upsampled and superimposed on the original image to see which part of the given image is activated by each prototype. As shown in the "Activation map" column in Figure 3 (left), the first prototype of the red-bellied woodpecker class activates most strongly on the head of the testing bird, and the second prototype on the wing: the most activated image patch of the given image for each prototype is marked by a bounding box in the "Original image" column – this is the image patch that the network considers to look like the corresponding prototype. In this case, our network finds a high similarity between the head of the given bird and the *prototypical* head of a red-bellied woodpecker (with a similarity score of $6.499$), as well as between the wing and the *prototypical* wing (with a similarity score of $4.392$). These similarity scores are weighted and summed together to give a final score for the bird belonging to this class. The reasoning process is similar for all other classes (Figure 3 (right)). The network finally correctly classifies the bird as a red-bellied woodpecker. Section S3 of the supplement provides more examples of how our ProtoPNet classifies previously unseen images of birds.

## 2.5 Comparison with baseline models and attention-based interpretable deep models

The accuracy of our ProtoPNet (with various base CNN architectures) on cropped bird images is compared to that of the corresponding baseline model in the top of Table 1: the first number in each cell gives the mean accuracy, and the second number gives the standard deviation, over three runs. To ensure fairness of comparison, the baseline models (without the prototype layer) were trained on the same augmented dataset of cropped bird images as the corresponding ProtoPNet. As we can see, the test accuracy of our ProtoPNet is comparable with that of the corresponding

Table 1: Top: Accuracy comparison on cropped bird images of CUB-200-2011
Bottom: Comparison of our model with other deep models

| Base | ProtoPNet | Baseline | Base | ProtoPNet | Baseline |
|------|-----------|----------|------|-----------|----------|
| VGG16 | $76.1 \pm 0.2$ | $74.6 \pm 0.2$ | VGG19 | $78.0 \pm 0.2$ | $75.1 \pm 0.4$ |
| Res34 | $79.2 \pm 0.1$ | $82.3 \pm 0.3$ | Res152 | $78.0 \pm 0.3$ | $81.5 \pm 0.4$ |
| Dense121 | $80.2 \pm 0.2$ | $80.5 \pm 0.1$ | Dense161 | $80.1 \pm 0.3$ | $82.2 \pm 0.2$ |

| Interpretability | Model: accuracy |
|------------------|-----------------|
| None | **B-CNN**[25]: 85.1 (bb), 84.1 (full) |
| Object-level attn. | **CAM**[56]: 70.5 (bb), 63.0 (full) |
| Part-level attention | **Part R-CNN**[53]: 76.4 (bb+anno.); **PS-CNN** [15]: 76.2 (bb+anno.); **PN-CNN** [3]: 85.4 (bb+anno.); **DeepLAC**[24]: 80.3 (anno.); **SPDA-CNN**[52]: 85.1 (bb+anno.); **PA-CNN**[19]: 82.8 (bb); **MG-CNN**[46]: 83.0 (bb), 81.7 (full); **ST-CNN**[16]: 84.1 (full); **2-level attn.**[49]: 77.9 (full); **FCAN**[26]: 82.0 (full); **Neural const.**[37]: 81.0 (full); **MA-CNN**[55]: 86.5 (full); **RA-CNN**[7]: 85.3 (full) |
| Part-level attn. + prototypical cases | **ProtoPNet** (ours): 80.8 (full, VGG19+Dense121+Dense161-based)<br>84.8 (bb, VGG19+ResNet34+DenseNet121-based) |

baseline (non-interpretable) model: the loss of accuracy is at most $3.5\%$ when we switch from the non-interpretable baseline model to our interpretable ProtoPNet. We can further improve the accuracy of ProtoPNet by *adding the logits of several ProtoPNet models together.* Since each ProtoPNet can be understood as a "scoring sheet" (as in Figure 3) for each class, adding the logits of several ProtoPNet models is equivalent to creating a combined scoring sheet where (weighted) similarity with prototypes from all these models is taken into account to compute the total points for each class – the combined model will have the same interpretable form when we combine several ProtoPNet models in this way, though there will be more prototypes for each class. The test accuracy on cropped bird images of combined ProtoPNets can reach $84.8\%$, which is on par with some of the best-performing deep models that were also trained on cropped images (see bottom of Table 1). We also trained a VGG19-, DenseNet121-, and DenseNet161-based ProtoPNet on full images: the test accuracy of the combined network can go above $80\%$ – at $80.8\%$, even though the test accuracy of each individual network is $72.7\%$, $74.4\%$, and $75.7\%$, respectively. Section S3.1 of the supplement illustrates how combining several ProtoPNet models can improve accuracy while preserving interpretability.

Moreover, our ProtoPNet provides a level of interpretability that is absent in other interpretable deep models. In terms of the type of explanations offered, Figure 4 provides a visual comparison of different types of model interpretability. At the coarsest level, there are models that offer object-level attention (e.g., class activation maps [56]) as explanation: this type of explanation (usually) highlights the entire object as the "reason" behind a classification decision, as shown in Figure 4(a). At a finer level, there are numerous models that offer part-level attention: this type of explanation highlights the important parts that lead to a classification decision, as shown in Figure 4(b). Almost all attention-based interpretable deep models offer this type of explanation (see the bottom of Table 1). In contrast, our model not only offers part-level attention, but also provides similar prototypical cases, and uses similarity to prototypical cases of a particular class as justification for classification (see Figure 4(c)). This type of interpretability is absent in other interpretable deep models. In terms of how attention is generated, some attention models generate attention with auxiliary part-localization models trained with part annotations (e.g., [53, 52, 3, 24, 15]); other attention models generate attention with "black-box" methods – e.g., RA-CNN [7] uses another neural network (attention proposal network) to decide where to look next; multi-attention CNN [55] uses aggregated convolutional feature maps as "part attentions." There is no explanation for why the attention proposal network decides to look at some region over others, or why certain parts are highlighted in those convolutional feature maps. In contrast, our ProtoPNet generates attention based on similarity with learned prototypes: it requires no part annotations for training, and explains its attention naturally – it is looking at *this* region of input because *this* region is similar to *that* prototypical example. Although other attention models focus on similar regions (e.g., head, wing, etc.) as our ProtoPNet, they cannot be made into a case-based reasoning model like ours: the only way to find prototypes on other attention models is to analyze *posthoc* what activates a convolutional filter of the model most strongly and think of that as a

prototype – however, since such prototypes do not participate in the actual model computation, any explanations produced this way are not always faithful to the classification decisions. The bottom of Table 1 compares the accuracy of our model with that of some state-of-the-art models on this dataset: "full" means that the model was trained and tested on full images, "bb" means that the model was trained and tested on images cropped using bounding boxes (or the model used bounding boxes in other ways), and "anno." means that the model was trained with keypoint annotations of bird parts. Even though there is some accuracy gap between our (combined) ProtoPNet model and the best of the state-of-the-art, this gap may be reduced through more extensive training effort, and the added interpretability in our model already makes it possible to bring richer explanations and better transparency to deep neural networks.

## 2.6 Analysis of latent space and prototype pruning

In this section, we analyze the structure of the latent space learned by our ProtoPNet. Figure 5(a) shows the three nearest prototypes to a test image of a Florida jay and of a cardinal. As we can see, the nearest prototypes for each of the two test images come from the same class as that of the image, and the test image's patch most activated by each prototype also corresponds to the same semantic concept as the prototype: in the case of the Florida jay, the most activated patch by each of the three nearest prototypes (all wing prototypes) indeed localizes the wing; in the case of the cardinal, the most activated patch by each of the three nearest prototypes (all head prototypes) indeed localizes the head. Figure 5(b) shows the nearest (i.e., most activated) image patches in the entire training/test set to three prototypes. As we can see, the nearest image patches to the first prototype in the figure are all heads of black-footed albatrosses, and the nearest image patches to the second prototype are all yellow stripes on the wings of golden-winged warblers. The nearest patches to the third prototype are feet of some gull. It is generally true that the nearest patches of a prototype all bear the same semantic concept, and they mostly come from those images in the same class as the prototype. Those prototypes whose nearest training patches have mixed class identities usually correspond to background patches, and they can be automatically pruned from our model. Section S8 of the supplement discusses pruning in greater detail.

# 3 Case study 2: car model identification

In this case study, we apply our method to car model identification. We trained our ProtoPNet on the Stanford Cars dataset [20] of 196 car models, using similar architectures and training algorithm as we did on the CUB-200-2011 dataset. The accuracy of our ProtoPNet and the corresponding baseline model on this dataset is reported in Section S6 of the supplement. We briefly state our performance here: the test accuracy of our ProtoPNet is comparable with that of the corresponding baseline model ($\leq 3\%$ difference), and that of a combined network of a VGG19-, ResNet34-, and DenseNet121-based ProtoPNet can reach $91.4\%$, which is on par with some state-of-the-art models on this dataset, such as B-CNN [25] ($91.3\%$), RA-CNN [7] ($92.5\%$), and MA-CNN [55] ($92.8\%$).

# 4 Conclusion

In this work, we have defined a form of interpretability in image processing (*this* looks like *that*) that agrees with the way humans describe their own reasoning in classification. We have presented ProtoPNet – a network architecture that accommodates this form of interpretability, described our specialized training algorithm, and applied our technique to bird species and car model identification.

**Supplementary Material and Code**: The supplementary material and code are available at `https://github.com/cfchen-duke/ProtoPNet`.

## Acknowledgments

This work was sponsored in part by a grant from MIT Lincoln Laboratory to C. Rudin.

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
