[Supplementary Material]

# *This* Looks Like *That*: Deep Learning for Interpretable Image Recognition (Supplementary Material)

**Chaofan Chen**[*]
Duke University
cfchen@cs.duke.edu

**Oscar Li**[*]
Duke University
oscarli@alumni.duke.edu

**Chaofan Tao**
Duke University
chaofan.tao@duke.edu

**Alina Jade Barnett**
Duke University
abarnett@cs.duke.edu

**Jonathan Su**
MIT Lincoln Laboratory[†]
su@ll.mit.edu

**Cynthia Rudin**
Duke University
cynthia@cs.duke.edu

## S1    Proof of Theorem 2.1

In this section, we provide a proof for Theorem 2.1 in the main paper.

We will introduce another notation, $\mathbf{p}_l^k$, for prototypes of a particular class: here, $k$ represents the class identity of the prototype and $l$ is the index of that prototype among all the prototypes of class $k$. In this way, the prototypes of class $k$ can be easily denoted as: $\mathbf{P}^k = \{\mathbf{p}_l^k\}_{l=1}^{m_k}$.

**Theorem 2.1** Let $h \circ g_{\mathbf{p}} \circ f$ be a ProtoPNet. For each $k$, $l$, we use $\mathbf{b}_l^k$ to denote the value of the $l$-th prototype for class $k$ **before** the projection of $\mathbf{p}_l^k$ to the nearest latent training patch of class $k$, and use $\mathbf{a}_l^k$ to denote its value **after** the projection. Let $\mathbf{x}$ be an input image that is correctly classified by the ProtoPNet before the projection, $\mathbf{z}_l^k = \arg\min_{\tilde{\mathbf{z}} \in \text{patches}(f(\mathbf{x}))} \|\tilde{\mathbf{z}} - \mathbf{b}_l^k\|_2$ be the nearest patch of $f(\mathbf{x})$ to the prototype $\mathbf{p}_l^k$ before the projection (i.e., $\mathbf{b}_l^k$), and $c$ be the correct class label of $\mathbf{x}$.

**Suppose** that:

   (A1) $\mathbf{z}_l^k$ is also the nearest latent patch to prototype $\mathbf{p}_l^k$ after the projection ($\mathbf{a}_l^k$), i.e., $\mathbf{z}_l^k = \arg\min_{\tilde{\mathbf{z}} \in \text{patches}(f(\mathbf{x}))} \|\tilde{\mathbf{z}} - \mathbf{a}_l^k\|_2$;

   (A2) there exists some $\delta$ with $0 < \delta < 1$ such that:

      (A2a) for all incorrect classes' prototypes $k \neq c$ and $l \in \{1, ..., m_k\}$, we have $\|\mathbf{a}_l^k - \mathbf{b}_l^k\|_2 \leq \theta \|\mathbf{z}_l^k - \mathbf{b}_l^k\|_2 - \sqrt{\epsilon}$, where we define $\theta = \min\left(\sqrt{1+\delta} - 1, 1 - \frac{1}{\sqrt{2-\delta}}\right)$ ($\epsilon$ comes from the prototype activation function $g_{\mathbf{p}_j}$ defined in Section 2.1);

      (A2b) for the correct class $c$ and for all $l \in \{1, ..., m_c\}$, we have $\|\mathbf{a}_l^c - \mathbf{b}_l^c\|_2 \leq (\sqrt{1+\delta} - 1)\|\mathbf{z}_l^c - \mathbf{b}_l^c\|_2$ and $\|\mathbf{z}_l^c - \mathbf{b}_l^c\|_2 \leq \sqrt{1-\delta}$;

   (A3) the number of prototypes is the same for each class, which we denote by $m'$.

---

[*]Contributed equally

[†]DISTRIBUTION STATEMENT A. Approved for public release. Distribution is unlimited. This material is based upon work supported by the Under Secretary of Defense for Research and Engineering under Air Force Contract No. FA8702-15-D-0001. Any opinions, findings, conclusions or recommendations expressed in this material are those of the author(s) and do not necessarily reflect the views of the Under Secretary of Defense for Research and Engineering.

**(A4)** for each class $k$, the weight connection in the fully connected last layer $h$ between a class $k$ prototype and the class $k$ logit is 1, and that between a non-class $k$ prototype and the class $k$ logit is 0 (i.e., $w_h^{(k,j)} = 1$ for all $j$ with $\mathbf{p}_j \in \mathbf{P}_k$ and $w_h^{(k,j)} = 0$ for all $j$ with $\mathbf{p}_j \notin \mathbf{P}_k$).

**Then** after projection, the output logit for the correct class $c$ can decrease at most by $\Delta_{\max} = m' \log((1+\delta)(2-\delta))$, and the output logit for every incorrect class $k \neq c$ can increase at most by $\Delta_{\max}$. If the output logits between the top-2 classes are at least $2\Delta_{\max}$ apart, then the projection of prototypes to their nearest latent training patches does not change the prediction of $\mathbf{x}$.

*Proof.* For any class $k$, let $L_k(\mathbf{x}, \{\mathbf{p}_l^k\}_{l=1}^{m'})$ denote its output logit for input image $x$ with the values of class $k$ prototypes being $\{p_l^k\}_{l=1}^{m'}$. By Assumption **(A4)**,

$$L_k(\mathbf{x}, \{\mathbf{p}_l^k\}_{l=1}^{m'}) = \sum_{l=1}^{m'} \log \left( \frac{\|\mathbf{z}_l^k - \mathbf{p}_l^k\|_2^2 + 1}{\|\mathbf{z}_l^k - \mathbf{p}_l^k\|_2^2 + \epsilon} \right).$$

Let $\Delta_k$ denote the change of the output logit of class $k$ as a result of the projection of prototypes $\{p_l^k\}_{l=1}^{m'}$ to their nearest latent training patches. This gives

$$\Delta_k = L_k(\mathbf{x}, \{\mathbf{a}_l^k\}_{l=1}^{m'}) - L_k(\mathbf{x}, \{\mathbf{b}_l^k\}_{l=1}^{m'})$$

$$= \sum_{l=1}^{m'} \left( \log \left( \frac{\|\mathbf{z}_l^k - \mathbf{a}_l^k\|_2^2 + 1}{\|\mathbf{z}_l^k - \mathbf{a}_l^k\|_2^2 + \epsilon} \right) - \log \left( \frac{\|\mathbf{z}_l^k - \mathbf{b}_l^k\|_2^2 + 1}{\|\mathbf{z}_l^k - \mathbf{b}_l^k\|_2^2 + \epsilon} \right) \right)$$

$$= \sum_{l=1}^{m'} \log \left( \frac{\|\mathbf{z}_l^k - \mathbf{a}_l^k\|_2^2 + 1}{\|\mathbf{z}_l^k - \mathbf{b}_l^k\|_2^2 + 1} \cdot \frac{\|\mathbf{z}_l^k - \mathbf{b}_l^k\|_2^2 + \epsilon}{\|\mathbf{z}_l^k - \mathbf{a}_l^k\|_2^2 + \epsilon} \right).$$

For each class $k \in \{1, ..., K\}$ and its prototypes $l \in \{1, ..., m'\}$, let

$$\Psi_l^k = \frac{\|\mathbf{z}_l^k - \mathbf{a}_l^k\|_2^2 + 1}{\|\mathbf{z}_l^k - \mathbf{b}_l^k\|_2^2 + 1} \cdot \frac{\|\mathbf{z}_l^k - \mathbf{b}_l^k\|_2^2 + \epsilon}{\|\mathbf{z}_l^k - \mathbf{a}_l^k\|_2^2 + \epsilon}.$$

(**Correct class**) We now derive a lower bound of $\Psi_l^c$ for the $l$-th prototype of the correct class $c$.

From the second inequality in **(A2b)**, we have

$$\frac{\|\mathbf{z}_l^c - \mathbf{a}_l^c\|_2^2 + 1}{\|\mathbf{z}_l^c - \mathbf{b}_l^c\|_2^2 + 1} \geq \frac{1}{\|\mathbf{z}_l^c - \mathbf{b}_l^c\|_2^2 + 1} \geq \frac{1}{2 - \delta}. \tag{1}$$

Now we want to lower-bound $\frac{\|\mathbf{z}_l^c - \mathbf{b}_l^c\|_2^2 + \epsilon}{\|\mathbf{z}_l^c - \mathbf{a}_l^c\|_2^2 + \epsilon}$ which is the second term in $\Psi_l^c$. We shall now prove

$$\frac{\|\mathbf{z}_l^c - \mathbf{b}_l^c\|_2^2 + \epsilon}{\|\mathbf{z}_l^c - \mathbf{a}_l^c\|_2^2 + \epsilon} \geq \frac{1}{1 + \delta}. \tag{2}$$

First, by the triangle inequality, we know $\|\mathbf{z}_l^c - \mathbf{a}_l^c\|_2 \leq \|\mathbf{z}_l^c - \mathbf{b}_l^c\|_2 + \|\mathbf{a}_l^c - \mathbf{b}_l^c\|_2$. As a result, we know

$$\frac{\|\mathbf{z}_l^c - \mathbf{b}_l^c\|_2^2 + \epsilon}{\|\mathbf{z}_l^c - \mathbf{a}_l^c\|_2^2 + \epsilon} \geq \frac{\|\mathbf{z}_l^c - \mathbf{b}_l^c\|_2^2 + \epsilon}{(\|\mathbf{z}_l^c - \mathbf{b}_l^c\|_2 + \|\mathbf{a}_l^c - \mathbf{b}_l^c\|_2)^2 + \epsilon}.$$

Then by **(A2b)**, we have

$$\|\mathbf{a}_l^c - \mathbf{b}_l^c\|_2 \leq (\sqrt{1 + \delta} - 1)\|\mathbf{z}_l^c - \mathbf{b}_l^c\|_2,$$

which implies

$$\|\mathbf{a}_l^c - \mathbf{b}_l^c\|_2 + \|\mathbf{z}_l^c - \mathbf{b}_l^c\|_2 \leq \sqrt{1 + \delta}\|\mathbf{z}_l^c - \mathbf{b}_l^c\|_2.$$

Squaring both sides of the above inequality, we have

$$(\|\mathbf{a}_l^c - \mathbf{b}_l^c\|_2 + \|\mathbf{z}_l^c - \mathbf{b}_l^c\|_2)^2 \leq (1 + \delta)\|\mathbf{z}_l^c - \mathbf{b}_l^c\|_2^2.$$

Adding $\epsilon$ to both sides of the equation, we obtain

$$(\|\mathbf{a}_l^c - \mathbf{b}_l^c\|_2 + \|\mathbf{z}_l^c - \mathbf{b}_l^c\|_2)^2 + \epsilon \leq (1+\delta)\|\mathbf{z}_l^c - \mathbf{b}_l^c\|_2^2 + \epsilon \leq (1+\delta)\|\mathbf{z}_l^c - \mathbf{b}_l^c\|_2^2 + (1+\delta)\epsilon.$$

Rearranging, we have

$$\frac{\|\mathbf{z}_l^c - \mathbf{b}_l^c\|_2^2 + \epsilon}{(\|\mathbf{z}_l^c - \mathbf{b}_l^c\|_2 + \|\mathbf{a}_l^c - \mathbf{b}_l^c\|_2)^2 + \epsilon} \geq \frac{1}{1+\delta}.$$

Now we obtain the desired result:

$$\frac{\|\mathbf{z}_l^c - \mathbf{b}_l^c\|_2^2 + \epsilon}{\|\mathbf{z}_l^c - \mathbf{a}_l^c\|_2^2 + \epsilon} \geq \frac{\|\mathbf{z}_l^c - \mathbf{b}_l^c\|_2^2 + \epsilon}{(\|\mathbf{z}_l^c - \mathbf{b}_l^c\|_2 + \|\mathbf{a}_l^c - \mathbf{b}_l^c\|_2)^2 + \epsilon} \geq \frac{1}{1+\delta}.$$

Combining inequalities (1) and (2), we have

$$\Psi_l^c = \frac{\|\mathbf{z}_l^c - \mathbf{a}_l^c\|_2^2 + 1}{\|\mathbf{z}_l^c - \mathbf{b}_l^c\|_2^2 + 1} \cdot \frac{\|\mathbf{z}_l^c - \mathbf{b}_l^c\|_2^2 + \epsilon}{\|\mathbf{z}_l^c - \mathbf{a}_l^c\|_2^2 + \epsilon} \geq \frac{1}{(1+\delta)(2-\delta)}.$$

This means that the change of the output logit of class $c$ as a result of the prototype projection $\{\mathbf{p}_l^c\}_{l=1}^{m'}$ to their nearest latent training patches satisfies

$$\Delta_c = \sum_{l=1}^{m'} \log \Psi_l^c \geq \sum_{l=1}^{m'} \log \left( \frac{1}{(1+\delta)(2-\delta)} \right) = m' \log \left( \frac{1}{(1+\delta)(2-\delta)} \right),$$

or equivalently,

$$-\Delta_c \leq -m' \log \left( \frac{1}{(1+\delta)(2-\delta)} \right) = m' \log((1+\delta)(2-\delta)).$$

This means that the worst decrease of the output logit of class $c$ as a result of prototype projection is $m' \log((1+\delta)(2-\delta))$, as desired.

(**Wrong Class**) We now derive an upper bound of $\Psi_{k,l}$ for the $l$-th prototype of any incorrect class $k \neq c$. We shall first give an upper bound of $\frac{\|\mathbf{z}_l^k - \mathbf{a}_l^k\|_2^2 + 1}{\|\mathbf{z}_l^k - \mathbf{b}_l^k\|_2^2 + 1}$. We show

$$\frac{\|\mathbf{z}_l^k - \mathbf{a}_l^k\|_2^2 + 1}{\|\mathbf{z}_l^k - \mathbf{b}_l^k\|_2^2 + 1} \leq 1 + \delta. \tag{3}$$

Using the triangle inequality, we obtain

$$\frac{\|\mathbf{z}_l^k - \mathbf{a}_l^k\|_2^2 + 1}{\|\mathbf{z}_l^k - \mathbf{b}_l^k\|_2^2 + 1} \leq \frac{(\|\mathbf{z}_l^k - \mathbf{b}_l^k\|_2 + \|\mathbf{a}_l^k - \mathbf{b}_l^k\|_2)^2 + 1}{\|\mathbf{z}_l^k - \mathbf{b}_l^k\|_2^2 + 1}. \tag{4}$$

By Assumption (**A2a**), we have

$$\|\mathbf{a}_l^k - \mathbf{b}_l^k\|_2 \leq (\sqrt{1+\delta} - 1)\|\mathbf{z}_l^k - \mathbf{b}_l^k\|_2 - \sqrt{\epsilon} \leq (\sqrt{1+\delta} - 1)\|\mathbf{z}_l^k - \mathbf{b}_l^k\|_2,$$

which then gives

$$\begin{aligned}
(\|\mathbf{z}_l^k - \mathbf{b}_l^k\|_2 + \|\mathbf{a}_l^k - \mathbf{b}_l^k\|_2)^2 &\leq (\|\mathbf{z}_l^k - \mathbf{b}_l^k\|_2 + (\sqrt{1+\delta} - 1)\|\mathbf{z}_l^k - \mathbf{b}_l^k\|_2)^2 \\
&= (\sqrt{1+\delta}\|\mathbf{z}_l^k - \mathbf{b}_l^k\|_2)^2 \\
&= (1+\delta)\|\mathbf{z}_l^k - \mathbf{b}_l^k\|_2^2.
\end{aligned} \tag{5}$$

Using the inequality (5), we obtain

$$\begin{aligned}
\frac{(\|\mathbf{z}_l^k - \mathbf{b}_l^k\|_2 + \|\mathbf{a}_l^k - \mathbf{b}_l^k\|_2)^2 + 1}{\|\mathbf{z}_l^k - \mathbf{b}_l^k\|_2^2 + 1} &\leq \frac{(1+\delta)\|\mathbf{z}_l^k - \mathbf{b}_l^k\|_2^2 + 1}{\|\mathbf{z}_l^k - \mathbf{b}_l^k\|_2^2 + 1} \\
&\leq \frac{(1+\delta)\|\mathbf{z}_l^k - \mathbf{b}_l^k\|_2^2 + 1 + \delta}{\|\mathbf{z}_l^k - \mathbf{b}_l^k\|_2^2 + 1} \\
&= 1 + \delta.
\end{aligned} \tag{6}$$

Combining inequalities (4) and (6), we have the desired result

$$\frac{\|\mathbf{z}_l^k - \mathbf{a}_l^k\|_2^2 + 1}{\|\mathbf{z}_l^k - \mathbf{b}_l^k\|_2^2 + 1} \le 1 + \delta.$$

Now we derive an upper bound for $\frac{\|\mathbf{z}_l^k - \mathbf{b}_l^k\|_2^2 + \epsilon}{\|\mathbf{z}_l^k - \mathbf{a}_l^k\|_2^2 + \epsilon}$. In particular, we show

$$\frac{\|\mathbf{z}_l^k - \mathbf{b}_l^k\|_2^2 + \epsilon}{\|\mathbf{z}_l^k - \mathbf{a}_l^k\|_2^2 + \epsilon} \le 2 - \delta. \tag{7}$$

By the triangle inequality, we have

$$\|\mathbf{z}_l^k - \mathbf{a}_l^k\|_2 \ge (\|\mathbf{z}_l^k - \mathbf{b}_l^k\|_2 - \|\mathbf{a}_l^k - \mathbf{b}_l^k\|_2).$$

Additionally, (**A2a**) implies $\|\mathbf{z}_l^k - \mathbf{b}_l^k\|_2 - \|\mathbf{a}_l^k - \mathbf{b}_l^k\|_2 > 0$, so we can square both sides of the above inequality and get:

$$\frac{\|\mathbf{z}_l^k - \mathbf{b}_l^k\|_2^2 + \epsilon}{\|\mathbf{z}_l^k - \mathbf{a}_l^k\|_2^2 + \epsilon} \le \frac{\|\mathbf{z}_l^k - \mathbf{b}_l^k\|_2^2 + \epsilon}{(\|\mathbf{z}_l^k - \mathbf{b}_l^k\|_2 - \|\mathbf{a}_l^k - \mathbf{b}_l^k\|_2)^2 + \epsilon} \le \left( \frac{\|\mathbf{z}_l^k - \mathbf{b}_l^k\|_2 + \sqrt{\epsilon}}{\|\mathbf{z}_l^k - \mathbf{b}_l^k\|_2 - \|\mathbf{a}_l^k - \mathbf{b}_l^k\|_2} \right)^2. \tag{8}$$

We now only need to upper bound $\frac{\|\mathbf{z}_l^k - \mathbf{b}_l^k\|_2 + \sqrt{\epsilon}}{\|\mathbf{z}_l^k - \mathbf{b}_l^k\|_2 - \|\mathbf{a}_l^k - \mathbf{b}_l^k\|_2}$. Again, using Assumption (**A2a**), we have

$$\|\mathbf{a}_l^k - \mathbf{b}_l^k\|_2 \le \left( 1 - \frac{1}{\sqrt{2 - \delta}} \right) \|\mathbf{z}_l^k - \mathbf{b}_l^k\|_2 - \sqrt{\epsilon}.$$

Rearranging, we have:

$$\frac{1}{\sqrt{2 - \delta}} \|\mathbf{z}_l^k - \mathbf{b}_l^k\|_2 + \sqrt{\epsilon} \le \|\mathbf{z}_l^k - \mathbf{b}_l^k\|_2 - \|\mathbf{a}_l^k - \mathbf{b}_l^k\|_2,$$

which leads us to conclude

$$\frac{1}{\sqrt{2 - \delta}} \|\mathbf{z}_l^k - \mathbf{b}_l^k\|_2 + \frac{\sqrt{\epsilon}}{\sqrt{2 - \delta}} \le \frac{1}{\sqrt{2 - \delta}} \|\mathbf{z}_l^k - \mathbf{b}_l^k\|_2 + \sqrt{\epsilon} \le \|\mathbf{z}_l^k - \mathbf{b}_l^k\|_2 - \|\mathbf{a}_l^k - \mathbf{b}_l^k\|_2.$$

The above inequality yields

$$\frac{\|\mathbf{z}_l^k - \mathbf{b}_l^k\|_2 + \sqrt{\epsilon}}{\|\mathbf{z}_l^k - \mathbf{b}_l^k\|_2 - \|\mathbf{a}_l^k - \mathbf{b}_l^k\|_2} \le \sqrt{2 - \delta}. \tag{9}$$

Combining inequalities (8) and (9), we establish the desired inequality

$$\frac{\|\mathbf{z}_l^k - \mathbf{b}_l^k\|_2^2 + \epsilon}{\|\mathbf{z}_l^k - \mathbf{a}_l^k\|_2^2 + \epsilon} \le (\sqrt{2 - \delta})^2 = 2 - \delta.$$

Inequalities (3) and (7) then give us

$$\Psi_l^k = \frac{\|\mathbf{z}_l^k - \mathbf{a}_l^k\|_2^2 + 1}{\|\mathbf{z}_l^k - \mathbf{b}_l^k\|_2^2 + 1} \cdot \frac{\|\mathbf{z}_l^k - \mathbf{b}_l^k\|_2^2 + \epsilon}{\|\mathbf{z}_l^k - \mathbf{a}_l^k\|_2^2 + \epsilon} \le (1 + \delta)(2 - \delta).$$

This means that the change of the output logit of class $k \ne c$ as a result of the projection of prototypes $\mathbf{p}_l^k$ to their nearest latent training patches satisfies

$$\Delta_k = \sum_{l=1}^{m'} \Psi_l^k \ge \sum_{l=1}^{m'} \log((1 + \delta)(2 - \delta)) = m' \log((1 + \delta)(2 - \delta)).$$

Since the increase of the output logit of class $k$ is exactly $\Delta_k$, we conclude that the worst increase of the output logit of class $k \ne c$ as a result of prototype projection is $m' \log((1 + \delta)(2 - \delta))$, as desired.

Finally, let $\Delta_{\max} = m' \log((1 + \delta)(2 - \delta))$. Suppose that the output logit $L_c(\mathbf{x}|\mathbf{b}_l^c)$ of the correct class $c$ **before** prototype projection is at least $2\Delta_{\max}$ higher than the output logit $L_k(\mathbf{x}|\mathbf{b}_l^k)$ of any other class $k \neq c$, i.e.,

$$L_c(\mathbf{x}, \{\mathbf{b}_l^c\}_{l=1}^{m'}) \geq L_k(\mathbf{x}, \{\mathbf{b}_l^k\}_{l=1}^{m'}) + 2\Delta_{max}, \quad \forall k \neq c$$

Since the output logit of the correct class $c$ satisfies

$$L_c(\mathbf{x}, \{\mathbf{a}_l^c\}_{l=1}^{m'}) \geq L_c(\mathbf{x}, \{\mathbf{b}_l^c\}_{l=1}^{m'}) - \Delta_{\max},$$

while the output logit of any incorrect class $k \neq c$ satifies

$$L_k(\mathbf{x}, \{\mathbf{a}_l^k\}_{l=1}^{m'}) \leq L_k(\mathbf{x}, \{\mathbf{b}_l^k\}_{l=1}^{m'}) + \Delta_{\max},$$

consequently, we have for any $k \neq c$,

$$
\begin{aligned}
L_c(\mathbf{x}, \{\mathbf{a}_l^c\}_{l=1}^{m'}) &\geq L_c(\mathbf{x}, \{\mathbf{b}_l^c\}_{l=1}^{m'}) - \Delta_{\max} \\
&\geq L_k(\mathbf{x}, \{\mathbf{b}_l^k\}_{l=1}^{m'}) + 2\Delta_{\max} - \Delta_{\max} \\
&= L_k(\mathbf{x}, \{\mathbf{b}_l^k\}_{l=1}^{m'}) + \Delta_{\max} \\
&\geq L_k(\mathbf{x}, \{\mathbf{a}_l^k\}_{l=1}^{m'}).
\end{aligned}
$$

Hence, in this case, the input image $\mathbf{x}$ will still be correctly classified as class $c$.

$\square$

## Interpretation of Theorem 2.1

Theorem 2.1 is presented in a general way in terms of the choice for $\delta$. However, to get a concrete feeling of the assumption, we can for simplicity set $\delta = \frac{9}{16}$.

Then (**A2a**) becomes $\|\mathbf{a}_l^k - \mathbf{b}_l^k\|_2 \leq (1 - \frac{4}{\sqrt{23}})\|\mathbf{z}_l^k - \mathbf{b}_l^k\|_2 - \sqrt{\epsilon}$. And $(1 - \frac{4}{\sqrt{23}}) \approx 0.17$, while

(**A2b**) becomes $\|\mathbf{a}_l^c - \mathbf{b}_l^c\|_2 \leq \frac{1}{4}\|\mathbf{z}_l^c - \mathbf{b}_l^c\|_2$ and $\|\mathbf{z}_l^c - \mathbf{b}_l^c\|_2 \leq \frac{\sqrt{7}}{4}$.

The requirement of $\|\mathbf{z}_l^c - \mathbf{b}_l^c\|_2 \leq \frac{\sqrt{7}}{4}$ is empirically always satisfied on our learned models. Regarding the relationship between $\|\mathbf{a}_l^c - \mathbf{b}_l^c\|_2$ and $\|\mathbf{z}_l^c - \mathbf{b}_l^c\|_2$, we can see that the requirement is tighter on the incorrect classes than the correct class. This is because for the wrong classes, we would expect that there exists no latent patch representation of the class-$c$ image $\mathbf{x}$ that is **very close** to non-class-$c$ prototypes. On the other hand, because our projection update pushes every non-class-$c$ prototype to the closest representation from its own class, the distance $\|\mathbf{a}_l^c - \mathbf{b}_l^c\|_2$ is generally much smaller than $\|\mathbf{z}_l^c - \mathbf{b}_l^c\|_2$. From this, we see that the assumptions made in the theorem are reasonable.

When the conditions are met, this theorem provably guarantees that the classifier's decision does not become worse on a large region in the image domain.

## S2   A probabilistic interpretation of ProtoPNet

In this section, we give a probabilistic interpretation of our ProtoPNet model's inference process.

We can think of the image classification problem as a conditional probability estimation problem, in which our goal is to estimate the conditional distribution $P(Y = k \mid \mathbf{X} = \mathbf{x})$, for all $k \in \{1, ..., K\}$, for all $\mathbf{x} \in \mathcal{X}$ ($\mathcal{X}$ is the image domain). We can rewrite this classification problem into a class-conditional density estimation problem for each class $k$ using **Bayes Theorem**:

$$P(Y = k \mid \mathbf{X} = \mathbf{x}) = \frac{P(\mathbf{X} = \mathbf{x} \mid Y = k)P(Y = k)}{\sum_{c=1}^{K} P(\mathbf{X} = \mathbf{x} \mid Y = c)P(Y = c)}.$$

However, learning a class-conditional density over the image space $\mathcal{X}$ for every class $k$ is a more daunting task than the classification problem. As a result, we make a reasonable assumption here

to shift the density estimation problem from the image space to a latent space. This will make the learning more tractable.

To introduce this assumption, we first provide our notations. We denote the domain of latent patches (equivalently, the domain of prototypes) to be $\Omega$. In this probabilistic derivation, we will treat the parameters $w_{\text{conv}}$ of our ProtoPNet's convolutional layers $f$ and the prototypes $\mathbf{P} = \bigcup_{k=1}^{K} \{\mathbf{p}_l^k\}_{l=1}^{m_k}$ as distribution parameters from a frequentist perspective – that is, these parameters will not be considered random. A more rigorous way to write the class-conditional probability would be: $P(\mathbf{X} = \mathbf{x} \mid Y = k; w_{\text{conv}}, \mathbf{P})$, but this formalism will be dropped as this dependence is ubiquitous and clear.

We now define a set of functions $\{f_l^k\}$, $f_l^k \colon \mathcal{X} \to \Omega$, where

$$\text{for all } \mathbf{x} \in \mathcal{X}, \ f_l^k(\mathbf{x}) := \arg \min_{\mathbf{z} \in \text{patches}(f(\mathbf{x}))} \|\mathbf{z} - \mathbf{p}_l^k\|_2.$$

We assume that for any image $\mathbf{x} \in \mathcal{X}$ and for any prototype $\mathbf{p}_l^k$, there exists only one latent patch $\mathbf{z} \in \text{patches}(f(\mathbf{x}))$ that is closest to the prototype $\mathbf{p}_l^k$ (in $L^2$-distance). Under this assumption, we can think of $f_l^k(\mathbf{x})$ as the (single) closest latent patch of the image $\mathbf{x}$ to the prototype $\mathbf{p}_l^k$, so that we clearly have $f_l^k(\mathbf{x}) \in \Omega$.

Since $w_{\text{conv}}$ and the prototypes $\mathbf{p}_l^k$ are deterministic parameters of the distribution, each function $f_l^k$ is deterministic and well-defined. We can now use $\mathbf{X}$ to denote the random variable over the image domain $\mathcal{X}$. Then $f_l^k(\mathbf{X})$, as a function of a random variable, is also a random variable.

We now write the class density over the image space as a product of two conditional probabilities:

$$
\begin{aligned}
&P(\mathbf{X} = \mathbf{x} \mid Y = k) \\
=&1 \cdot P(\mathbf{X} = \mathbf{x} \mid Y = k) \\
=&P(f_1^k(\mathbf{X}) = f_1^k(\mathbf{x}), ..., f_{m_k}^k(\mathbf{X}) = f_{m_k}^k(\mathbf{x}) \mid \mathbf{X} = \mathbf{x}, Y = k) \cdot P(\mathbf{X} = \mathbf{x} \mid Y = k) \\
=&P(f_1^k(\mathbf{X}) = f_1^k(\mathbf{x}), ..., f_{m_k}^k(\mathbf{X}) = f_{m_k}^k(\mathbf{x}), \mathbf{X} = \mathbf{x} \mid Y = k) \\
=&P(\mathbf{X}=\mathbf{x}|f_1^k(\mathbf{X})=f_1^k(\mathbf{x}),...,f_{m_k}^k(\mathbf{X})=f_{m_k}^k(\mathbf{x}),Y=k)\cdot P(f_1^k(\mathbf{X})=f_1^k(\mathbf{x}),...,f_{m_k}^k(\mathbf{X})=f_{m_k}^k(\mathbf{x})|Y=k).
\end{aligned}
$$

The derivation from the second to the third line uses the fact that if we know the value of $\mathbf{X}$ being $\mathbf{x}$, then the values of $f_l^k(\mathbf{X})$ must be $f_l^k(\mathbf{x})$ with probability of 1. Now we are ready to introduce our assumption to simplify the density estimation:

**Assumption (i)**

$$
\begin{aligned}
&\forall \mathbf{x} \in \mathcal{X}, \ \ \forall a, b \in \{1, ..., K\}, \\
&P(\mathbf{X} = \mathbf{x} \mid f_1^a(\mathbf{X}) = f_1^a(\mathbf{x}), ..., f_{m_a}^a(\mathbf{X}) = f_{m_a}^a(\mathbf{x}), Y = a) \\
&= P(\mathbf{X} = \mathbf{x} \mid f_1^b(\mathbf{X}) = f_1^b(\mathbf{x}), ..., f_{m_b}^b(\mathbf{X}) = f_{m_b}^b(\mathbf{x}), Y = b).
\end{aligned}
$$

This assumption says that for any given image $\mathbf{x}$, the probability of $\mathbf{X}$ being of value $\mathbf{x}$ given all of its closest latent patches to prototypes of class $k$ and the fact that the image is indeed of class $k$ is the same for every single class $k \in \{1, ..., K\}$. We can roughly think of this assumption as: for any class, knowing an unknown image's closest latent patches to prototypes of that class and the fact that unknown image actually belongs to that class gives us **the same level of uncertainty** about the true value of the image.

A more restrictive but simpler case of Assumption (i) is to believe **in additional** that:

$$
\begin{aligned}
&P(\mathbf{X} = \mathbf{x} \mid f_1^c(\mathbf{X}) = f_1^c(\mathbf{x}), ..., f_{m_c}^c(\mathbf{X}) = f_{m_c}^c(\mathbf{x}), Y = c) = \\
&\qquad P(\mathbf{X} = \mathbf{x} \mid f_1^c(\mathbf{X}) = f_1^c(\mathbf{x}), ..., f_{m_c}^c(\mathbf{X}) = f_{m_c}^c(\mathbf{x})), \\
&\qquad\qquad\qquad\qquad\qquad\qquad\qquad \forall c \in \{1, ..., K\}.
\end{aligned}
$$

This additional assumption means that the closest latent patches to prototypes of class $c$ gives us enough information about the image's pixel values, so that the additional knowledge of the image's actual class $c$ does not change our uncertainty about those pixel values. Then Assumption (i) becomes the same as saying:

$$
\begin{aligned}
&P(\mathbf{X} = \mathbf{x} \mid f_1^a(\mathbf{X}) = f_1^a(\mathbf{x}), ..., f_{m_a}^a(\mathbf{X}) = f_{m_a}^a(\mathbf{x})) = \\
&\qquad P(\mathbf{X} = \mathbf{x} \mid f_1^b(\mathbf{X}) = f_1^b(\mathbf{x}), ..., f_{m_b}^b(\mathbf{X}) = f_{m_b}^b(\mathbf{x})), \\
&\qquad\qquad\qquad\qquad\qquad\qquad\qquad \forall a, b \in \{1, ..., K\}.
\end{aligned}
$$

Combining Assumption (i) with Bayes Theorem, the class-conditional densities over the image space $\mathcal{X}$ can now be reduced to the class-conditional densities over the space $\Omega^{m_k}$ (the Cartesian product of the space of latent patches):

$$P(Y = k \mid \mathbf{X} = \mathbf{x}) = \frac{P(f_1^k(\mathbf{X}) = f_1^k(\mathbf{x}), ..., f_{m_k}^k(\mathbf{X}) = f_{m_k}^k(\mathbf{x}) \mid Y = k) \cdot P(Y = k)}{\sum_{c=1}^{K} P(f_1^c(\mathbf{X}) = f_1^c(\mathbf{x}), ..., f_{M_c}^c(\mathbf{X}) = f_{M_c}^c(\mathbf{x}) \mid Y = c) \cdot P(Y = c)}.$$

Alternatively, we can write

$$\forall k \in \{1, ..., K\},$$
$$P(Y = k \mid \mathbf{X} = \mathbf{x}) \propto P(f_1^k(\mathbf{X}) = f_1^k(\mathbf{x}), ..., f_{m_k}^k(\mathbf{X}) = f_{m_k}^k(\mathbf{x}) \mid Y = k) \cdot P(Y = k).$$

To simplify the joint distribution of random variables $f_1^k(\mathbf{X}), f_2^k(\mathbf{X}), ..., f_{m_k}^k(\mathbf{X})$ over $\Omega^{m_k}$, we make the second assumption in our derivation:

**Assumption (ii)**

for all $\mathbf{x} \in \mathcal{X}$,

$$P(f_1^k(\mathbf{X}) = f_1^k(\mathbf{x}), ..., f_{m_k}^k(\mathbf{X}) = f_{m_k}^k(\mathbf{x}) \mid Y = k) = \prod_{l=1}^{m_k} P(f_l^k(\mathbf{X}) = f_l^k(\mathbf{x}) \mid Y = k).$$

This assumption is not claiming that the random variables $\{f_l^k(\mathbf{X})\}$ are independent over the entire space of $\Omega^{m_k}$ – these are only independent in the subspace $\{(\mathbf{z}_1^k, ..., \mathbf{z}_{m_k}^k) \in \Omega^{m_k}$ : there exists $\mathbf{x} \in \mathcal{X}$ such that $\mathbf{z}_1^k = f_1^k(\mathbf{x}), ..., \mathbf{z}_{m_k}^k = f_{m_k}^k(\mathbf{x})\}$. Assumption (ii) allows us to factorize the joint distribution of $\{f_l^k(\mathbf{X})\}$ into a product of class-conditional densities of individual random variables.

Now, we have

$$P(Y = k \mid \mathbf{X} = \mathbf{x})$$
$$= \frac{P(f_1^k(\mathbf{X}) = f_1^k(\mathbf{x}), f_2^k(\mathbf{X}) = f_2^k(\mathbf{x}), ..., f_{m_k}^k(\mathbf{X}) = f_{m_k}^k(\mathbf{x}) \mid Y = k) \cdot P(Y = k)}{\sum_{c=1}^{K} P(f_1^c(\mathbf{X}) = f_1^c(\mathbf{x}), f_2^c(\mathbf{X}) = f_2^c(\mathbf{x}), ..., f_{m_c}^c(\mathbf{X}) = f_{m_c}^c(\mathbf{x}) \mid Y = c) \cdot P(Y = c)}$$
$$= \frac{\left[\prod_{l=1}^{m_k} P(f_l^k(\mathbf{X}) = f_l^k(\mathbf{x}) \mid Y = k)\right] \cdot P(Y = k)}{\sum_{c=1}^{K} \left\{\left[\prod_{l=1}^{m_c} P(f_l^c(\mathbf{X}) = f_l^c(\mathbf{x}) \mid Y = c)\right] \cdot P(Y = c)\right\}}.$$

Alternatively, we can write:

$$\text{for all } k \in \{1, ..., K\},$$
$$P(Y = k \mid \mathbf{X} = \mathbf{x}) \propto \left[\prod_{l=1}^{m_k} P(f_l^k(\mathbf{X}) = f_l^k(\mathbf{x}) \mid Y = k)\right] \cdot P(Y = k).$$

To use case-based reasoning in our model, we want our model to predict high probability for a particular class $k$ when it finds part(s) of the image to be semantically similar to some prototypical part(s) of class $k$ but almost no part of the image to be similar to prototypical parts of any other class. The latent space of the model should ideally have the property that semantically similar image parts will be close to each other (in $L^2$-distance) in the latent space. As a result, we naturally want the density of $f_l^k(\mathbf{X})$ in the latent space $\Omega$ to satisfy

$$P(f_l^k(\mathbf{X}) = z \mid Y = k) = d_l^k(\|\mathbf{z} - \mathbf{p}_l^k\|_2),$$

where $d_l^k : [0, \infty) \to [0, \infty)$ is monotonically decreasing and satisfies $\int_\Omega d_l^k(\|z - \mathbf{p}_l^k\|_2)dz = 1$. By this requirement, the latent distribution of random variable $f_l^k(\mathbf{X})$ is spherically symmetrical and has its mode at the value of the prototype $\mathbf{p}_l^k$.

Plugging the individual distribution into the class probability prediction, we have

$$P(Y = k \mid \mathbf{X} = \mathbf{x}) = \frac{\left[\prod_{j=1}^{m_k} d_l^k(\|f_l^k(\mathbf{x}) - \mathbf{p}_l^k\|_2)\right] \cdot P(Y = k)}{\sum_{c=1}^{K} \left\{ \left[\prod_{j=1}^{m_c} d_l^c(\|f_l^c(\mathbf{x}) - p_l^c\|_2)\right] \cdot P(Y = c) \right\}}. \tag{10}$$

Now that the probabilistic derivation of our model is complete, it remains to show that our current implementation can be interpreted as implementing this framework.

When the weight matrix of our ProtoPNet's (fully-connected) last layer has value $1$ between the prototypes and the classes they represent, and value $0$ everywhere else (which is approximately true after convex optimization of the last layer), the probability prediction by the ProtoPNet for class $k$ is as follows:

$$P(Y = k \mid \mathbf{X} = \mathbf{x}) = \frac{\exp\left(\sum_{l=1}^{m_k} \log\left(1 + \frac{1 - \epsilon}{\|f_l^k(\mathbf{x}) - \mathbf{p}_l^k\|_2^2 + \epsilon}\right)\right)}{\sum_{c=1}^{K} \exp\left(\sum_{l=1}^{m_c} \log\left(1 + \frac{1 - \epsilon}{\|f_l^c(\mathbf{x}) - \mathbf{p}_l^c\|_2^2 + \epsilon}\right)\right)}.$$

Simplifying, we have

$$P(Y = k \mid \mathbf{X} = \mathbf{x}) = \frac{\prod_{l=1}^{m_k} \frac{\|f_l^k(\mathbf{x}) - \mathbf{p}_l^k\|_2^2 + 1}{\|f_l^k(\mathbf{x}) - \mathbf{p}_l^k\|_2^2 + \epsilon}}{\sum_{c=1}^{K} \prod_{l=1}^{m_c} \frac{\|f_l^c(\mathbf{x}) - \mathbf{p}_l^c\|_2^2 + 1}{\|f_l^c(\mathbf{x}) - \mathbf{p}_l^c\|_2^2 + \epsilon}}. \tag{11}$$

Comparing Equations (10) and (11), we see that for our current model, we have: (1) $\Omega = [0, 1]^{H_1 \times W_1 \times D}$; (2) $P(Y = c) = \frac{1}{K}$, for all $c \in \{1, ..., K\}$; and (3) $d_l^k(r) = C_l^k \cdot \frac{r^2 + 1}{r^2 + \epsilon}$. The constant $C_l^k$ ensures that the integration over the latent domain $\Omega$ gives value of $1$. When the number of prototypes for every class is the same, the $\prod_{l=1}^{m_k} C_l^k$ will approximately cancel out in the numerator and denominator of Equation 10, leaving us with the current expression we use for our classification model.

**Remark** As we see from the above derivation, the inference of our ProtoPNet can be interpreted in a probabilistic framework with reasonable assumptions. In fact, the framework we developed opens up new potential choices of similiarity functions within our prototype layer. If we let $\Omega = \mathbb{R}^{H_1 \times W_1 \times D}$, we can for example use the Gaussian distribution for the class-conditional marginal distribution of $f_l^k(\mathbf{X})$ given $Y = k$, which in turns means that the prototype activation function will be $g_{\mathbf{p}_j}(\mathbf{z}) = \max_{\tilde{\mathbf{z}} \in \text{patches}(\mathbf{z})} -\|\tilde{\mathbf{z}} - \mathbf{p}_j\|_2^2$. The empirical evaluation of the performance of prototype activation functions inspired by this probabilistic framework is left as future work.

In addition, learning the prototypes $\mathbf{p}_l^k$ through stochastic gradient descent before projection can be understood as learning the parametric modes for $f_l^k(\mathbf{X})$'s distribution in the space of latent patches, and prototype projection can be understood as moving the mode of the distribution to the closest latent patch from the training set. The effect of projection will be minimal if the distribution mode does not move much as a result of prototype projection.

Furthermore, adding the logits of several ProtoPNet models for final prediction can also be understood through this probabilistic interpretation. The addition of the logits can be understood as the product of the class-conditional marginal distributions of different models' $f_l^k(\mathbf{X})$ given $Y = k$. Generally, including more random variables $f_l^k(\mathbf{X})$ in the joint density estimation improves the predictive performance of our model.

Why is this bird classfied as a Baltimore oriole?

Evidence for this bird being a Baltimore oriole:

| Original image (box showing part that looks like prototype) | Prototype | Training image where prototype comes from | Activation map | Similarity score | Class connection | Points contributed |
|---|---|---|---|---|---|---|
| | | | | 3.738 | × 1.065 | = 3.981 |
| | | | | 3.489 | × 1.242 | = 4.333 |
| | | | | 3.055 | × 1.061 | = 3.241 |
| ⋮ | ⋮ | ⋮ | ⋮ | ⋮ | | ⋮ |

Total points to Baltimore oriole: 23.919

Evidence for this bird being a hooded oriole:

| Original image (box showing part that looks like prototype) | Prototype | Training image where prototype comes from | Activation map | Similarity score | Class connection | Points contributed |
|---|---|---|---|---|---|---|
| | | | | 3.504 | × 1.212 | = 4.247 |
| | | | | 1.586 | × 1.131 | = 1.794 |
| | | | | 1.137 | × 1.058 | = 1.203 |
| ⋮ | ⋮ | ⋮ | ⋮ | ⋮ | | ⋮ |

Total points to hooded oriole: 12.307

(a) VGG16-based ProtoPNet.

Why is this bird classfied as a Baltimore oriole?

Evidence for this bird being a Baltimore oriole:

| Original image (box showing part that looks like prototype) | Prototype | Training image where prototype comes from | Activation map | Similarity score | Class connection | Points contributed |
|---|---|---|---|---|---|---|
| | | | | 3.428 | × 1.344 | = 4.607 |
| | | | | 2.114 | × 1.290 | = 2.727 |
| | | | | 1.806 | × 1.343 | = 2.425 |
| ⋮ | ⋮ | ⋮ | ⋮ | ⋮ | | ⋮ |

Total points to Baltimore oriole: 15.211

Evidence for this bird being a hooded oriole:

| Original image (box showing part that looks like prototype) | Prototype | Training image where prototype comes from | Activation map | Similarity score | Class connection | Points contributed |
|---|---|---|---|---|---|---|
| | | | | 4.011 | × 1.314 | = 5.270 |
| | | | | 0.820 | × 1.527 | = 1.252 |
| | | | | 0.789 | × 1.514 | = 1.195 |
| ⋮ | ⋮ | ⋮ | ⋮ | ⋮ | | ⋮ |

Total points to hooded oriole: 10.865

(b) VGG19-based ProtoPNet.

Why is this bird classfied as a Baltimore oriole?

Evidence for this bird being a Baltimore oriole:

| Original image (box showing part that looks like prototype) | Prototype | Training image where prototype comes from | Activation map | Similarity score | Class connection | Points contributed |
|---|---|---|---|---|---|---|
| | | | | 3.085 | × 0.923 | = 2.847 |
| | | | | 2.922 | × 1.022 | = 2.986 |
| | | | | 2.789 | × 0.987 | = 2.753 |
| ⋮ | ⋮ | ⋮ | ⋮ | ⋮ | | ⋮ |

Total points to Baltimore oriole: 18.446

Evidence for this bird being hooded oriole:

| Original image (box showing part that looks like prototype) | Prototype | Training image where prototype comes from | Activation map | Similarity score | Class connection | Points contributed |
|---|---|---|---|---|---|---|
| | | | | 1.253 | × 1.052 | = 1.318 |
| | | | | 0.859 | × 1.075 | = 0.923 |
| | | | | 0.795 | × 1.102 | = 0.876 |
| ⋮ | ⋮ | ⋮ | ⋮ | ⋮ | | ⋮ |

Total points to hooded oriole: 6.920

(c) ResNet34-based ProtoPNet.

# S3 More examples of how our ProtoPNet classifies birds

In this section, we provide more examples of how our ProtoPNet classifies previously unseen images of birds.

Figures 1 and 2 provide two examples of how our ProtoPNet (with various base architectures) correctly classifies a previously unseen image of a bird and how our network explains its prediction. In each of these figures, the left side presents evidence for the given bird belonging to the class with the highest logit, and the right side presents evidence for the given bird belonging to a closely related class. We shall give some general observations regarding the ways in which our network thinks that

(d) ResNet152-based ProtoPNet.

(e) DenseNet121-based ProtoPNet.

(f) DenseNet161-based ProtoPNet.

Figure 1: How our ProtoPNet correctly classifies an image of Baltimore oriole.

Why is this bird classfied as a pied-billed grebe?

Evidence for this bird being a pied-billed grebe:

| Original image (box showing part that looks like prototype) | Prototype | Training image where prototype comes from | Activation map | Similarity score | Class connection | Points contributed |
|---|---|---|---|---|---|---|
| | | | | 4.159 | × 0.930 = | 3.868 |
| | | | | 3.315 | × 0.872 = | 2.891 |
| | | | | 2.999 | × 1.029 = | 3.086 |
| ⋮ | ⋮ | ⋮ | ⋮ | ⋮ | ⋮ | ⋮ |

Total points to pied-billed grebe: 19.204

Evidence for this bird being a horned grebe:

| Original image (box showing part that looks like prototype) | Prototype | Training image where prototype comes from | Activation map | Similarity score | Class connection | Points contributed |
|---|---|---|---|---|---|---|
| | | | | 1.922 | × 1.086 = | 2.087 |
| | | | | 1.749 | × 1.198 = | 2.095 |
| | | | | 1.571 | × 0.946 = | 1.486 |
| ⋮ | ⋮ | ⋮ | ⋮ | ⋮ | ⋮ | ⋮ |

Total points to horned grebe: 12.158

(a) VGG16-based ProtoPNet.

Why is this bird classfied as a pied-billed grebe?

Evidence for this bird being a pied-billed grebe:

| Original image (box showing part that looks like prototype) | Prototype | Training image where prototype comes from | Activation map | Similarity score | Class connection | Points contributed |
|---|---|---|---|---|---|---|
| | | | | 4.150 | × 1.295 = | 5.374 |
| | | | | 3.088 | × 1.243 = | 3.838 |
| | | | | 2.985 | × 1.002 = | 2.991 |
| ⋮ | ⋮ | ⋮ | ⋮ | ⋮ | ⋮ | ⋮ |

Total points to pied-billed grebe: 23.826

Evidence for this bird being a horned grebe:

| Original image (box showing part that looks like prototype) | Prototype | Training image where prototype comes from | Activation map | Similarity score | Class connection | Points contributed |
|---|---|---|---|---|---|---|
| | | | | 2.171 | × 1.468 = | 2.565 |
| | | | | 1.120 | × 1.457 = | 2.318 |
| | | | | 1.032 | × 1.436 = | 2.079 |
| ⋮ | ⋮ | ⋮ | ⋮ | ⋮ | ⋮ | ⋮ |

Total points to horned grebe: 11.225

(b) VGG19-based ProtoPNet.

Why is this bird classfied as a pied-billed grebe?

Evidence for this bird being a pied-billed grebe:

| Original image (box showing part that looks like prototype) | Prototype | Training image where prototype comes from | Activation map | Similarity score | Class connection | Points contributed |
|---|---|---|---|---|---|---|
| | | | | 2.705 | × 1.189 = | 3.216 |
| | | | | 2.523 | × 1.150 = | 2.901 |
| | | | | 2.356 | × 1.060 = | 2.497 |
| ⋮ | ⋮ | ⋮ | ⋮ | ⋮ | ⋮ | ⋮ |

Total points to pied-billed grebe: 18.967

Evidence for this bird being a horned grebe:

| Original image (box showing part that looks like prototype) | Prototype | Training image where prototype comes from | Activation map | Similarity score | Class connection | Points contributed |
|---|---|---|---|---|---|---|
| | | | | 0.652 | × 1.043 = | 0.680 |
| | | | | 0.579 | × 1.018 = | 0.589 |
| | | | | 0.458 | × 1.040 = | 0.476 |
| ⋮ | ⋮ | ⋮ | ⋮ | ⋮ | ⋮ | ⋮ |

Total points to horned grebe: 3.962

(c) ResNet34-based ProtoPNet.

Why is this bird classified as a pied-billed grebe?

Evidence for this bird being a pied-billed grebe:

| Original image (box showing part that looks like prototype) | Prototype | Training image where prototype comes from | Activation map | Similarity score | Class connection | | Points contributed |
|---|---|---|---|---|---|---|---|
| | | | | 7.548 | × | 1.142 = | 8.620 |
| | | | | 4.761 | × | 1.147 = | 5.461 |
| | | | | 3.082 | × | 1.122 = | 3.458 |
| ⋮ | ⋮ | ⋮ | ⋮ | ⋮ | | ⋮ | ⋮ |
| | | Total points to pied-billed grebe: | | | | | 27.830 |

Evidence for this bird being a horned grebe:

| Original image (box showing part that looks like prototype) | Prototype | Training image where prototype comes from | Activation map | Similarity score | Class connection | | Points contributed |
|---|---|---|---|---|---|---|---|
| | | | | 0.670 | × | 1.239 = | 0.830 |
| | | | | 0.474 | × | 1.287 = | 0.610 |
| | | | | 0.470 | × | 1.226 = | 0.576 |
| ⋮ | ⋮ | ⋮ | ⋮ | ⋮ | | ⋮ | ⋮ |
| | | Total points to horned grebe: | | | | | 4.991 |

(d) ResNet152-based ProtoPNet.

Why is this bird classified as a pied-billed grebe?

Evidence for this bird being a pied-billed grebe:

| Original image (box showing part that looks like prototype) | Prototype | Training image where prototype comes from | Activation map | Similarity score | Class connection | | Points contributed |
|---|---|---|---|---|---|---|---|
| | | | | 6.912 | × | 0.735 = | 5.080 |
| | | | | 5.999 | × | 0.686 = | 4.115 |
| | | | | 2.634 | × | 1.274 = | 3.356 |
| ⋮ | ⋮ | ⋮ | ⋮ | ⋮ | | ⋮ | ⋮ |
| | | Total points to pied-billed grebe: | | | | | 23.261 |

Evidence for this bird being a horned grebe:

| Original image (box showing part that looks like prototype) | Prototype | Training image where prototype comes from | Activation map | Similarity score | Class connection | | Points contributed |
|---|---|---|---|---|---|---|---|
| | | | | 2.150 | × | 1.096 = | 2.356 |
| | | | | 2.052 | × | 1.108 = | 2.274 |
| | | | | 1.821 | × | 1.330 = | 2.422 |
| ⋮ | ⋮ | ⋮ | ⋮ | ⋮ | | ⋮ | ⋮ |
| | | Total points to horned grebe: | | | | | 14.154 |

(e) DenseNet121-based ProtoPNet.

Why is this bird classified as a pied-billed grebe?

Evidence for this bird being a pied-billed grebe:

| Original image (box showing part that looks like prototype) | Prototype | Training image where prototype comes from | Activation map | Similarity score | Class connection | | Points contributed |
|---|---|---|---|---|---|---|---|
| | | | | 5.256 | × | 0.884 = | 4.646 |
| | | | | 4.466 | × | 0.860 = | 3.841 |
| | | | | 4.108 | × | 0.899 = | 3.693 |
| ⋮ | ⋮ | ⋮ | ⋮ | ⋮ | | ⋮ | ⋮ |
| | | Total points to pied-billed grebe: | | | | | 30.276 |

Evidence for this bird being a horned grebe:

| Original image (box showing part that looks like prototype) | Prototype | Training image where prototype comes from | Activation map | Similarity score | Class connection | | Points contributed |
|---|---|---|---|---|---|---|---|
| | | | | 2.190 | × | 1.113 = | 2.437 |
| | | | | 1.857 | × | 1.050 = | 1.950 |
| | | | | 1.684 | × | 1.100 = | 1.852 |
| ⋮ | ⋮ | ⋮ | ⋮ | ⋮ | | ⋮ | ⋮ |
| | | Total points to horned grebe: | | | | | 12.671 |

(f) DenseNet161-based ProtoPNet.

Figure 2: How our ProtoPNet correctly classifies an image of pied-billed grebe.

Why is this bird classfied as a Wilson's warbler?

Evidence for this bird being a Wilson's warbler:

| Original image (box showing part that looks like prototype) | Prototype | Training image where prototype comes from | Activation map | Similarity score | Class connection | Points contributed |
|---|---|---|---|---|---|---|
| | | | | 3.341 | × 1.443 | = 4.821 |
| | | | | 3.302 | × 1.450 | = 4.788 |
| | | | | 2.159 | × 1.442 | = 3.113 |
| ⋮ | ⋮ | ⋮ | ⋮ | ⋮ | ⋮ | ⋮ |

Total points to Wilson's warbler: 19.473

Evidence for this bird being a prothonotary warbler:

| Original image (box showing part that looks like prototype) | Prototype | Training image where prototype comes from | Activation map | Similarity score | Class connection | Points contributed |
|---|---|---|---|---|---|---|
| | | | | 1.722 | × 1.105 | = 1.903 |
| | | | | 1.626 | × 1.085 | = 1.764 |
| | | | | 1.605 | × 1.173 | = 1.883 |
| ⋮ | ⋮ | ⋮ | ⋮ | ⋮ | ⋮ | ⋮ |

Total points to prothonotary warbler: 10.234

(a) VGG16-based ProtoPNet.

Why is this bird classfied as a Wilson's warbler?

Evidence for this bird being a Wilson's warbler:

| Original image (box showing part that looks like prototype) | Prototype | Training image where prototype comes from | Activation map | Similarity score | Class connection | Points contributed |
|---|---|---|---|---|---|---|
| | | | | 3.226 | × 1.394 | = 4.497 |
| | | | | 2.097 | × 1.414 | = 2.965 |
| | | | | 1.996 | × 1.319 | = 2.633 |
| ⋮ | ⋮ | ⋮ | ⋮ | ⋮ | ⋮ | ⋮ |

Total points to Wilson's warbler: 16.996

Evidence for this bird being a prothonotary warbler:

| Original image (box showing part that looks like prototype) | Prototype | Training image where prototype comes from | Activation map | Similarity score | Class connection | Points contributed |
|---|---|---|---|---|---|---|
| | | | | 2.727 | × 1.359 | = 3.706 |
| | | | | 2.422 | × 1.455 | = 3.524 |
| | | | | 2.286 | × 1.388 | = 3.173 |
| ⋮ | ⋮ | ⋮ | ⋮ | ⋮ | ⋮ | ⋮ |

Total points to prothonotary warbler: 15.876

(b) VGG19-based ProtoPNet.

Why is this bird classfied as a Wilson's warbler?

Evidence for this bird being a Wilson's warbler:

| Original image (box showing part that looks like prototype) | Prototype | Training image where prototype comes from | Activation map | Similarity score | Class connection | Points contributed |
|---|---|---|---|---|---|---|
| | | | | 1.208 | × 1.418 | = 1.713 |
| | | | | 1.042 | × 1.449 | = 1.510 |
| | | | | 0.873 | × 1.426 | = 1.245 |
| ⋮ | ⋮ | ⋮ | ⋮ | ⋮ | ⋮ | ⋮ |

Total points to Wilson's warbler: 9.836

Evidence for this bird being a prothonotary warbler:

| Original image (box showing part that looks like prototype) | Prototype | Training image where prototype comes from | Activation map | Similarity score | Class connection | Points contributed |
|---|---|---|---|---|---|---|
| | | | | 0.763 | × 0.911 | = 0.695 |
| | | | | 0.763 | × 0.911 | = 0.695 |
| | | | | 0.759 | × 0.943 | = 0.716 |
| ⋮ | ⋮ | ⋮ | ⋮ | ⋮ | ⋮ | ⋮ |

Total points to prothonotary warbler: 5.462

(c) ResNet34-based ProtoPNet.

Why is this bird classfied as a Wilson's warbler?

Evidence for this bird being a Wilson's warbler:

| Original image (box showing part that looks like prototype) | Prototype | Training image where prototype comes from | Activation map | Similarity score | | Class connection | | Points contributed |
|---|---|---|---|---|---|---|---|---|
| | | | | 1.300 | × | 1.053 | = | 1.369 |
| | | | | 1.300 | × | 1.053 | = | 1.369 |
| | | | | 1.189 | × | 1.104 | = | 1.313 |
| ⋮ | ⋮ | ⋮ | ⋮ | ⋮ | | ⋮ | | ⋮ |

Total points to Wilson's warbler: 8.337

Evidence for this bird being a prothonotary warbler:

| Original image (box showing part that looks like prototype) | Prototype | Training image where prototype comes from | Activation map | Similarity score | | Class connection | | Points contributed |
|---|---|---|---|---|---|---|---|---|
| | | | | 1.070 | × | 1.051 | = | 1.125 |
| | | | | 0.983 | × | 1.055 | = | 1.037 |
| | | | | 0.973 | × | 1.050 | = | 1.022 |
| ⋮ | ⋮ | ⋮ | ⋮ | ⋮ | | ⋮ | | ⋮ |

Total points to prothonotary warbler: 6.618

(d) ResNet152-based ProtoPNet.

Why is this bird classfied as a Wilson's warbler?

Evidence for this bird being a Wilson's warbler:

| Original image (box showing part that looks like prototype) | Prototype | Training image where prototype comes from | Activation map | Similarity score | | Class connection | | Points contributed |
|---|---|---|---|---|---|---|---|---|
| | | | | 5.356 | × | 1.039 | = | 5.565 |
| | | | | 5.356 | × | 1.042 | = | 5.581 |
| | | | | 3.589 | × | 1.228 | = | 4.407 |
| ⋮ | ⋮ | ⋮ | ⋮ | ⋮ | | ⋮ | | ⋮ |

Total points to Wilson's warbler: 27.189

Evidence for this bird being a prothonotary warbler:

| Original image (box showing part that looks like prototype) | Prototype | Training image where prototype comes from | Activation map | Similarity score | | Class connection | | Points contributed |
|---|---|---|---|---|---|---|---|---|
| | | | | 1.784 | × | 1.284 | = | 2.291 |
| | | | | 1.692 | × | 1.274 | = | 2.156 |
| | | | | 1.543 | × | 1.223 | = | 1.887 |
| ⋮ | ⋮ | ⋮ | ⋮ | ⋮ | | ⋮ | | ⋮ |

Total points to prothonotary warbler: 14.609

(e) DenseNet121-based ProtoPNet.

Why is this bird incorrectly classified as a prothonotary warbler, instead of a Wilson's warbler?

Evidence for this bird being a Wilson's warbler:

| Original image (box showing part that looks like prototype) | Prototype | Training image where prototype comes from | Activation map | Similarity score | | Class connection | | Points contributed |
|---|---|---|---|---|---|---|---|---|
| | | | | 1.342 | × | 1.357 | = | 1.821 |
| | | | | 1.189 | × | 1.247 | = | 1.483 |
| | | | | 1.189 | × | 1.247 | = | 1.483 |
| ⋮ | ⋮ | ⋮ | ⋮ | ⋮ | | ⋮ | | ⋮ |

Total points to Wilson's warbler: 9.744

Evidence for this bird being a prothonotary warbler:

| Original image (box showing part that looks like prototype) | Prototype | Training image where prototype comes from | Activation map | Similarity score | | Class connection | | Points contributed |
|---|---|---|---|---|---|---|---|---|
| | | | | 2.951 | × | 1.125 | = | 3.320 |
| | | | | 2.401 | × | 1.140 | = | 2.737 |
| | | | | 1.636 | × | 1.209 | = | 1.978 |
| ⋮ | ⋮ | ⋮ | ⋮ | ⋮ | | ⋮ | | ⋮ |

Total points to prothonotary warbler: 12.391

(f) DenseNet161-based ProtoPNet: this network mistakes the Wilson's warbler as a prothonotary warbler – the total points to Wilson's warbler is less than that to prothonotary warbler.

Figure 3: How our ProtoPNet (correctly and incorrectly) classifies an image of Wilson's warbler.

the given image is similar to the prototypical cases. The detailed reasoning process of our network has been explained in our main paper, and will not be repeated here.

Figure 1 demonstrates how our ProtoPNet (with various base architectures) correctly classifies an image of a Baltimore oriole. In particular, every ProtoPNet is able to learn the prototypical golden chest/abdomen of a Baltimore oriole, and is able to associate the golden chest/abdomen of the (previously unseen) given image to the prototypical golden chest (or abdomen) of a Baltimore oriole: for each of those prototypes that correspond to the characteristic golden chest/abdomen of a Baltimore oriole, our network is able to pick out, on the previously unseen given image, a similar patch with a golden chest/abdomen and highlight the golden chest/abdomen on the previously unseen bird in its prototype activation map (e.g., the first prototype in Figure 1e (left), which corresponds to the characteristic golden chest of a Baltimore oriole, identifies and highlights the golden chest on the previously unseen bird). On the other hand, some of our networks also think that the the golden abdomen of the given bird looks like the prototypical golden abdomen of a hooded oriole (e.g., the first prototype in Figure 1a (left), which corresponds to the characteristic golden abdomen of a hooded oriole, identifies and highlights the golden abdomen on the given bird). It is worth pointing out that sometimes a network can "mistakenly" believe that a certain pattern in the given image looks like a prototypical part of some class of birds: for example, the second and the third prototype in Figure 1b (left) "mistakenly" think that the wing and the chest of the given bird looks like the prototypical head of a hooded oriole – this is, however, not too surprising because the orange-black coloration on the wing and the chest of the given bird does look somewhat like the same coloration on the prototypical head of a hooded oriole; however, such remote resemblance is reflected in our network by very small similarity scores between the given image and those two prototypes (0.820 and 0.789), which (fortunately) means that the network does not believe that the aforementioned similarity is strong. Not surprisingly, when our ProtoPNet accumulates the evidence presented by the comparison with all the prototypes, it sees that the evidence for the given bird being a Baltimore oriole is the strongest, and concludes that the bird is a Baltimore oriole.

Note that sometimes a prototype can be duplicated in our network (e.g., the first and the second prototype in Figure 1d are the same): this results from the projection of each prototype onto the closest latent representation of training image patches from the prototype's designated class (described in Section 2.2: Training algorithm in the main paper) – in this case, the closest training patches to both prototypes are the same before the projection stage, and consequently both prototypes are projected onto the same patch in the latent space. This means that some of the learned prototypes in our network are repeated. However, this is not a problem because we can conceptually understand the repeated prototypes as one prototype, with its weight connection to each class in the fully connected last layer being the sum of the weight connections of those repeated prototypes to that class. Thus, we can understand the first and the second prototype in Figure 1d as one Baltimore oriole prototype with class connection $0.808 + 0.808 = 1.616$. This also means that the actual number of prototypes used by our ProtoPNet is in general *less* than the pre-determined number of prototypes when the network architecture is specified.

Figure 2 demonstrates how our ProtoPNet correctly classifies a previously unseen image of a pied-billed grebe. In particular, every ProtoPNet is able to learn the prototypical head of a pied-billed grebe, and is able to associate the head of the given bird to the prototypical head of a pied-billed grebe: for each of those prototypes that correspond to the head of a pied-billed grebe, our network is able to pick out, on the previously unseen image, a similar patch that contains the head of the given bird, and also highlight the head of the given bird in its prototype activation map – this strong resemblance is also reflected in high similarity scores between the given image and those prototypes. This shows that our network thinks that the head of the given bird looks like the prototypical head of a pied-billed grebe. On the other hand, each of our networks also thinks that there is some resemblance between the neck/back of the given bird and that of a horned grebe, but such resemblance is not very strong.

### S3.1  How model combination can improve accuracy

Figures 3a through 3e demonstrates how our ProtoPNet based on VGG16, VGG19, ResNet34, ResNet152, and DenseNet121 correctly classifies an image of a Wilson's warbler. On the other hand, Figure 3f shows why the DenseNet161-based ProtoPNet misclassifies the given bird as a prothonotary warbler instead of its true identity – a Wilson's warbler. As we can see in Figure 3f (right), the

DenseNet161-based ProtoPNet (mistakenly) thinks that the head of the given bird is more similar to the prototypical head of a prothonotary warbler than to that of a Wilson's warbler: this is, for example, shown by the first prothonotary warbler prototype (in 3f (right)) having a higher similarity score than the first Wilson's warbler prototype (in 3f (left)). In the end, this network finds more evidence for the given bird being a prothonotary warbler than being a Wilson warbler, and misclassifies the bird as a prothonotary warbler.

Here we illustrate why combining several ProtoPNet models is a good idea: it can improve prediction accuracy while preserving interpretability. For simplicity, suppose that we combine the VGG16-based and the DenseNet161-based ProtoPNet together, by adding the logits (i.e., the total points to different classes) together. When the combined model makes a prediction on the Wilson's warbler, the total points to Wilson's warbler becomes $19.473 + 9.744 = 29.217$, while the total points to prothonotary warbler becomes $10.234 + 12.391 = 22.625$. Thus, the combined network will correctly classify the image as a Wilson's warbler. At the same time, adding the logits (the total points to different classes) together in this way **preserves the interpretability** of our ProtoPNet model, because it is conceptually equivalent to having a scoring sheet that makes comparison with more prototypes (in different latent spaces) and accumulating all the evidence together into the total points for each class.

## S4    More examples of nearest prototypes of given images

In this section, we provide more examples of the nearest prototypes of given test images.

Figure 4 shows the three nearest prototypes to each of the five test bird images, from different ProtoPNet models (with various base architectures): for each ProtoPNet, the three nearest prototypes to a given test bird image are displayed, with prototypical parts shown in boxes, on the top row, and the same test image with the patch closest to each prototype shown in a bounding box, is displayed below the corresponding prototype. For a given image, we define its nearest prototype as the one that forms the closest patch-prototype pair in the latent space, over all latent patches of the given image. As we can see from Figure 4, the nearest prototypes for each of these test images generally come from the same class as that of the image, and the patch that is closest to (i.e., most activated by) each prototype also corresponds to the same semantic concept. For example, Figure 4a shows the three nearest prototypes to a test image of a rose-breasted grosbeak, from different ProtoPNet models (with various base architectures): as we can see, the three nearest prototypes from each of the ProtoPNet models indeed all come from the rose-breasted grosbeak class, and moreover, the nearest prototype from each of the ProtoPNet models corresponds to the rose breast characteristic of the species, and the closest (i.e., most activated) patch to the prototype indeed localizes the rose breast of the given bird. There are some exceptions: for example, the third nearest prototype from the VGG16-based ProtoPNet for the Lincoln sparrow corresponds to the wing of a western meadowlark (see Figure 4c. This is understandable, because a Lincoln sparrow has wing stripes much like those of a western meadowlark. Hence, it is not too surprising that the wing of the Lincoln sparrow in Figure 4c is fairly close to the prototypical wing of a western meadowlark. This shows that the latent space learned by our ProtoPNet does have a clustering structure, where semantically similar patches that are relevant for classification are clustered together.

## S5    More examples of nearest image patches of given prototypes

Figure 5 shows the nearest training and test image patches of three prototypes from different ProtoPNet models (with various base architectures). The prototypes are displayed with prototypical parts shown in bounding boxes, and the nearest training and test images to each prototype are displayed with the patch closest to that prototype in a bounding box. As we can see, the nearest image patches to each prototype in the figure all localize the same semantic part as the prototypical part of that prototype, and it is generally true that the nearest patches of a prototype mostly come from those images in the same class as that of the prototype.

(a) Nearest prototypes of the rose-breasted grosbeak from ProtoPNet models with various base architectures.

(b) Nearest prototypes of the ringed kingfisher from ProtoPNet models with various base architectures.

(c) Nearest prototypes of the Lincoln sparrow from ProtoPNet models with various base architectures.

(d) Nearest prototypes of the Kentucky warbler from ProtoPNet models with various base architectures.

(e) Nearest prototypes of the Cerulean warbler from ProtoPNet models with various base architectures.

Figure 4: Nearest prototypes of five test images: in each group of images, the three nearest prototypes of the corresponding test image are displayed, with prototypical parts shown in boxes, on the top row, and the same test image with the patch closest to each prototype shown in a bounding box, is displayed below the corresponding prototype.

## S6 Accuracy on Stanford Cars, an example of how our ProtoPNet classifies a car and nearest image patches of car prototypes

In this section, we compare the accuracy of our ProtoPNet with that of the corresponding baseline model on the Stanford Cars dataset (see Table 1: the first number in each cell gives the mean accuracy, and the second number gives the standard deviation, over three runs). As we can see, the test accuracy of our ProtoPNet is comparable with that of the corresponding baseline model, and the loss of accuracy is within 3% when we switch from the non-interpretable baseline model to our interpretable ProtoPNet. The test accuracy of a combined network of the three ProtoPNets in Table 1 can reach 91.4%, which is on par with some state-of-the-art models on this dataset, such as B-CNN [2] (91.3%), RA-CNN [1] (92.5%), and MA-CNN [3] (92.8%).

Table 1: Accuracy comparison on Stanford Cars

| Baseline architecture | Accuracy of ProtoPNet | Accuracy of baseline |
|---|---|---|
| VGG19 | $87.4 \pm 0.3$ | $85.9 \pm 0.2$ |
| ResNet34 | $86.1 \pm 0.1$ | $85.4 \pm 0.1$ |
| DenseNet121 | $86.8 \pm 0.1$ | $89.7 \pm 0.1$ |

(a) Nearest image patches to prototypes from VGG16-based ProtoPNet.

(b) Nearest image patches to prototypes from VGG19-based ProtoPNet.

(c) Nearest image patches to prototypes from ResNet34-based ProtoPNet.

(d) Nearest image patches to prototypes from ResNet152-based ProtoPNet.

(e) Nearest image patches to prototypes from DenseNet121-based ProtoPNet.

(f) Nearest image patches to prototypes from DenseNet161-based ProtoPNet.

Figure 5: Nearest (most activated) image patches to prototypes.

Figure 6: How our ProtoPNet correctly classifies an image of 2012 Honda Accord coupe.

Figure 7: Nearest image patches to prototypes from ProtoPNet trained on Stanford Cars.

We also provide an example of how our (VGG19-based) ProtoPNet trained on Stanford Cars classifies a previously unseen image of a 2012 Honda Accord coupe (Figure 6). In this particular example, our network thinks that the front of the car in the given image looks a lot like the prototypical front (the first two prototypes include the car logo) of a 2012 Honda Accord coupe, and it looks somewhat like the prototypical front of a Toyota Camry sedan, but the similarity between the front of the given car and the prototypical front of a 2012 Honda Accord coupe is stronger, than that between the front of the given car and the prototypical front of a 2012 Toyota Camry sedan.

Figure 7 provide some examples of prototypes from our (VGG19-based) ProtoPNet model trained on Stanford Cars, along with the nearest image patches to those prototypes from both the training and the test set.

Figure 8: How to visualize a prototype.

## S7 Detailed description of prototype visualization

In this section, we provide a detailed explanation of how we visualize prototypes of a trained ProtoPNet model. Recall that after the projection of prototypes to the closest latent patch of some training image, a prototype $\mathbf{p}_j$ is exactly equal to some patch of the latent representation $f(\mathbf{x})$, of some training image $\mathbf{x}$. Since the patch of $\mathbf{x}$ that corresponds to the prototype $\mathbf{p}_j$ should be the one that the prototype $\mathbf{p}_j$ activates the most strongly on, we visualize the prototype $\mathbf{p}_j$ by first obtaining the activation map of $\mathbf{x}$ by the prototype $\mathbf{p}_j$: this can be done by forwarding $\mathbf{x}$ through the trained ProtoPNet model and upsampling the activation map produced by the prototype unit $g_{\mathbf{p}_j}$ to the size of the image $\mathbf{x}$ (Step (1) in Figure 8). After we obtain such an activation map, we can locate the patch of $\mathbf{x}$ on which $\mathbf{p}_j$ has the strongest activation by finding the high activation region in the (upsampled) activation map (Step (2) in Figure 8). In our experiments, we define the high activation region in an upsampled activation map as the smallest rectangular region that encloses pixels whose corresponding activation value in the aforementioned activation map is at least $95\%$-percentile of all activation values in that same map. Finally, we can visualize the prototype $\mathbf{p}_j$ using the image patch of $\mathbf{x}$ that corresponds to the high activation region (Step (3) in Figure 8).

## S8 Detailed description of prototype pruning

Recall from our analysis of the nearest image patches of given prototypes, that it is generally true that the nearest (i.e., most activated) patches of a prototype mostly come from those images in the same class as that of the prototype. However, there are exceptions to this general observation. In our experiments, we find that for some prototypes in a trained ProtoPNet model, the nearest (training or test) image patches can all come from different classes. This is often the case when the prototype corresponds to a "background" patch (e.g., a patch of the sky). This happens during the training process because a background prototype can often be useful in distinguishing different species of birds: for example, a prototype that corresponds to a patch of water can be useful in distinguishing water birds from others.

In this section, we describe an algorithm for prototype pruning, that can reduce the number of prototypes for each class, and at the same time, remove "background" prototypes automatically from the reasoning process of our ProtoPNet. The pruning algorithm starts by first finding the $k$-nearest latent patches of training images to each prototype $\mathbf{p}_j$. Since we know the labels of all training images, we know the class labels of the $k$ training images where the $k$-nearest latent patches to $\mathbf{p}_j$ come from. Out of these $k$ training images, if less than $\tau$ (a predefined pruning threshold) of them come from the designated class of the prototype $\mathbf{p}_j$, then we assume that the prototype $\mathbf{p}_j$ most likely corresponds to some background patch and we remove the prototype $\mathbf{p}_j$ (along with its last layer connections) from the ProtoPNet model. After pruning, we can again optimize the last layer (see Section 2.2: Training algorithm) to boost accuracy further.

In our experiments, we set $k = 6$ and $\tau = 3$. Table 2 shows the effect of pruning (and subsequent optimization of the last layer) on ProtoPNet models trained on cropped bird images of CUB-200-2011, as well as the number of prototypes pruned (recall that we used 10 prototypes per class, so there were a total of 2000 prototypes for 200 classes before pruning). As we can see from Table 2, pruning has little effect on the accuracy of our ProtoPNet, if it is followed by the optimization of the last fully connected layer.

Figure 9 shows some examples of pruned prototypes and their nearest image patches from the training and the test set. As we can see, the first pruned prototype in Figure 9 is a prototypical tree branch, the second some prototypical background color, and the third corresponds to an image patch of sky.

Table 2: Effect of pruning (and subsequent optimization of the last layer) on various ProtoPNet models trained on cropped bird images of CUB-200-2011. The number of prototypes before pruning was 2000 for each model.

| Base architecture of ProtoPNet model | Accuracy before pruning | Accuracy after pruning | Accuracy after pruning and optimizing last layer | Number of prototypes pruned $(k = 6, \tau = 3)$ |
|---|---|---|---|---|
| VGG16 | 76.3 | 71.8 | 76.0 | 651 |
| VGG19 | 78.2 | 74.4 | 78.0 | 666 |
| ResNet34 | 79.2 | 79.1 | 79.5 | 345 |
| ResNet152 | 78.3 | 78.1 | 78.6 | 266 |
| DenseNet121 | 80.4 | 77.0 | 79.2 | 524 |
| DenseNet161 | 80.1 | 78.3 | 79.9 | 473 |

Figure 9: Examples of pruned prototypes and their nearest image patches from the training and the test set.

## S9 Implementation and training details

In this section, we describe the data augmentation techniques we used, as well as our choice of hyperparameters and training details.

### S9.1 Data augmentation

In our experiments on CUB-200-2011, since the dataset has only about 30 images per class, we performed offline data augmentation using random rotation, skew, shear, distortion, and left-right flip to enlarge the training set, so that each class has approximately 1200 training images.

In our experiments on Stanford Cars, since the dataset has only about 40 images per class, we performed offline data augmentation using random rotation, skew, shear, and left-right flip to enlarge the training set, so that each class has approximately 1300 training images.

### S9.2 Architecture and hyperparameter choices

In our experiments on CUB-200-2011, we used the convolutional layers from VGG-16, VGG-19, ResNet-34, ResNet-152, DenseNet-121, and DenseNet-161 (initialized with filters pretrained on ImageNet), followed by two additional $1 \times 1$ convolutional layers, as the convolutional part of our ProtoPNet. The number of output channels in each of the two additional convolutional layers is chosen to be the same as the number of channels in a prototype. For each base architecture, we chose from three possible values: 128, 256, 512, for the number of channels in a prototype (using cross validation): for VGG-16, VGG-19, DenseNet-121, DenseNet-161, we used 128 as the number of channels in a prototype; for ResNet-34, we used 256 as the number of channels in a prototype; for

```
1  initialize t₀ ← 0; w_base ← weights pre-trained on ImageNet; w_add ← Kaiming uniform initialization (He et al., 2015);
2       ∀j: prototype p_j ← Uniform([0,1]^{H₁×W₁×D}); ∀k,j: w_h^{(k,j)} ← 1 if p_j ∈ P_k, w_h^{(k,j)} ← 0 if p_j ∉ P_k;
3  while NOT(converge AND Clst < −Sep) do
      /* Stage 1:  SGD of layers before the last */
4    for SGD training epoch t = t₀ + 1, ..., t₀ + N_SGD do
5      foreach batch [X̄, Ȳ] from [X, Y] do
6        if t > 5 then            /* Pretrained weights and biases are fixed during the warm-up period */
7          w_base ← w_base − η_base^{(t)} ∇_{w_base} L(X̄, Ȳ);
8        w_add ← w_add − η_add^{(t)} ∇_{w_add} L(X̄, Ȳ); P ← P − η_p^{(t)} ∇_P L(X̄, Ȳ);
9    t₀ ← t₀ + N_SGD;
      /* Stage 2:  projection of prototypes */
10   foreach prototype p_j do
11     k ← class of p_j; p_j ← arg min_{z∈{z̃:z̃∈patches(f(x)) ∀(x,y)∈[X,Y] s.t. y=k}} ‖z − p_j‖₂;
      /* Stage 3:  convex optimization of last layer */
12   for convex training epoch t′ = 1, ..., N_convex do
13     foreach batch [X̄, Ȳ] from [X, Y] do w_h ← w_h − η_convex ∇_{w_h} L_convex(X̄, Ȳ) ;
```

Figure 10: Overview of training algorithm.

ResNet-152, we used 512 as the number of channels in a prototype. We used $1 \times 1$ as the spatial dimension of each prototype (i.e., $H_1 = 1$ and $W_1 = 1$): given that the spatial dimension of the convolutional output for a $224 \times 224$ image is only $7 \times 7$, a $1 \times 1$ prototype is already large enough to represent a significant part of the original image in the pixel space (we want to learn prototypes **focused on specific parts**). The number of prototypes can be chosen with prior domain knowledge or hyperparameter search: we used 10 prototypes per class, because CUB-200-2011 provides (at most) 15 part locations per image, so we believe that 10 prototypes per class should be enough to capture a variety of bird parts. Note that the part locations (keypoint annotations) provided with the dataset were not used by our algorithm during training – we used only image-level labels.

In our experiments on Stanford Cars, we used the convolutional layers from VGG-19, ResNet-34, and DenseNet-121 (initialized with filters pretrained on ImageNet), followed by two additional $1 \times 1$ convolutional layers, as the convolutional part of our ProtoPNet. We chose the number of output channels in the additional convolutional layers, the number of channels in a prototype, and the spatial dimension of each prototype, in the same way as we did on CUB-200-2011. We again used 10 prototypes per class, because we believe that 10 prototypes should be enough to capture different views of a car.

### S9.3 Overview of training algorithm

In the algorithm chart in Figure 10: $w_{\text{base}}$ and $w_{\text{add}}$ denote the parameters of the base and additional convolutional layers; $N_{\text{SGD}}$ and $N_{\text{convex}}$ denote the number of training epochs in stage 1 and 3; $L$ and $L_{\text{convex}}$ denote the loss function of stage 1 and 3; $\eta_{\text{base}}^{(t)}$, $\eta_{\text{add}}^{(t)}$, $\eta_{\text{p}}^{(t)}$, $\eta_{\text{convex}}$ are learning rates ($t$ denotes epoch number). The choice of learning rates, as well as the coefficients of the terms in the loss function, is discussed in the next section (Section S9.4).

### S9.4 Training parameters

In our experiments on both CUB-200-2011 and Stanford Cars, we set the coefficient of the cluster cost to 0.8, and the coefficient of the separation cost to 0.08 during stochastic gradient descent of layers before the last layer, and we set the coefficient of the $L^1$-regularization term (on the weight connection between each prototype of class $k$ and the logit of class $k' \neq k$) to $10^{-4}$ during convex optimization of the last layer. For the coefficient of the cluster cost and the coefficient of the separation cost, we considered three different settings: $(1, 0.1)$, $(0.8, 0.08)$, $(0.6, 0.06)$, and chose the pair $(0.8, 0.08)$ using cross validation. For the coefficient of the $L^1$-regularization term, we considered $10^{-3}$, $10^{-4}$, and $10^{-5}$, and chose $10^{-4}$ also by cross validation.

In our experiments, we started our training with a "warm-up" stage, in which we loaded and froze the pre-trained weights and biases, and focused on training the two additional convolutional layers and the prototype layer (without requiring each prototype to be exactly some latent training patch) for 5 epochs. The learning rate we used in this sub-stage is $3 \times 10^{-3}$. Afterward, we trained all the convolutional layers and the prototype layer jointly, using $10^{-4}$ learning rate for those layers that were pretrained on ImageNet, and $3 \times 10^{-3}$ learning rate for the two additional convolutional

layers and the prototype layer. We reduced the learning rate by a factor of $0.1$ every $5$ epochs, and we performed prototype projection and convex optimization of the last layer (for $20$ iterations) every two times we reduced the learning rate. We stopped training when training accuracy converged and the cluster cost became smaller than the separation cost on the training set.

### S9.5 Training software and platform

We implemented our ProtoPNet using PyTorch. The experiments were run on $4$ NVIDIA Tesla P100 GPUs or $8$ NVIDIA Tesla K80 GPUs.

Our code is available at `https://github.com/cfchen-duke/ProtoPNet`.

### S9.6 Links to datasets

CUB-200-2011 can be downloaded from:
`http://www.vision.caltech.edu/visipedia/CUB-200-2011.html`

Stanford Cars can be downloaded from:
`https://ai.stanford.edu/~jkrause/cars/car_dataset.html`