[Reviews · NeurIPS 2019]

Reviewer 1



The prototypical parts network presented in this work is original and potentially very useful learning framework for domains where process-based interpretability is critical. The method is thoroughly evaluated against alternative approaches and performs comparable to other state-of-the-art interpretable learning algorithms. The paper is well written, well motivated, and is accompanied by empirical results to validate the algorithmic contributions. Overall, I would recommend this paper for acceptance. One place for improvement is the discussion of this work in the context of alternative interpretable approaches, specifically the methods that show comparable accuracy. The authors briefly mention the advantage of having parts based interpretability. However, I would appreciate a longer discussion of the difference between this method and, for example, the RA-CNN which performs slightly better. This discussion would provide additional useful context to the prototypical parts network as well as provide an opportunity to discuss tradeoffs of different approaches. Another place for improvement would bee more thorough exploration of the ProtoPNet architecture and hyper-parameter choices. For example, what is the effect of changing the number of prototypes per class? Do the similarity scores (across prototypes) relate in a meaningful way to the confidence of the final class prediction? I also wonder if there are some domains where finding prototypical images could be a useful goal in and of itself? Curious what the authors think about this direction. ========== response to rebuttal ========== After reading the thorough author response I am have increased my score. I think this paper is a clear accept now.

Reviewer 2



The idea of learning prototypes to improve interpretation is interesting and the comparison with prototypical cases provides clear explanation of the classification result. The presentation of the paper is easy to follow. However, I still have some concerns/questions regarding the method and the experiments. 1. The choice of prototypes P seems essential for the performance. It would better if some more implementation details can be provided. For example, 1) how are P initialized? 2) How to choose H_1 and W_1, since the similarity is based on the L2 distance, the size of the prototypes may be important. 3) Is the update of P (between line 183 and 184) performed every iteration? 4) How to choose m_k? I believe some of the above question might be answered by carefully checking the code provided by the authors. Yet I think it would be helpful to summarize these in the paper, maybe using an algorithm summarization, for readers to get a better overview. 2. The comparison between training latent patches and the prototypes, as well as the update of prototypes may increase the computational cost since the algorithm needs to go over all possible patches. What is the time complexity? 3. The interpretation results seem a bit weak to me, as only Figure 4 provides some comparison with related interpretation methods on one example. =========================== I like the idea of this paper, and the author response has addressed my concerns in complexity, and comparison with other methods. I have changed my score to 6 accordingly.

Reviewer 3



The paper is well written, the problem is well motivated, and references are sufficient. I don't see any major problems. Lines 47-54 present important related work. Did the authors try to compare their results with the results of attention models? My understanding is that attention models would provide regions that the classifier is looking at. Would those methods provide more accurate, or similar, or the same regions? I understand that the main point of ProtoPNet is to identify prototypical cases, but perhaps the regions identified by the attention models would be equally good? Also, what is the problem with the existing attention methods that would not allow one to extract prototypical cases? Is there anything wrong with those architectures that would not allow for that? It would be useful if the authors could explain that. What is the impact of m_k (the number of prototypes for each class k)? Did the authors try smaller/larger values of these parameters? How was the current value of 10 determined? Humans probably look at fewer than 10 prototypical regions in their identification tasks. How important was the L^2 distance in this algorithm? Did the authors try any other metrics? How was L^2 selected?

[Author Response · NeurIPS 2019]

**More comparison with related interpretable methods (R1, R2, R3):** In our paper, we discussed the main difference between our ProtoPNet and related attention models in terms of the **type of explanations** offered: **our ProtoPNet not only offers attention on several parts** (akin to attention models), **but also provides similar prototypical cases to those parts** (which attention models cannot provide) **as built-in justification for classification**. In terms of **how attention is generated**: **some attention models generate attention with auxiliary part-localization models trained with part annotations** (e.g. part-based R-CNN (Zhang et al., ECCV 2014), SPDA-CNN (Zhang et al., CVPR 2016), pose-normalized CNN (Branson et al., 2014), DeepLAC (Lin et al., CVPR 2015), part-stacked CNN (Huang et al., CVPR 2016)); **other attention models generate attention with "black-box" methods** – e.g. RA-CNN (Fu et al., CVPR 2017) uses another neural network (attention proposal network) to decide where to look next; multi-attention CNN (Zheng et al., ICCV 2017) uses aggregated conv-feature maps as "part attentions." There is **no explanation** for why the attention proposal network decides to look at some region over others, or why certain parts are highlighted in those conv-feature maps. In contrast, **our ProtoPNet generates attention based on similarity with learned prototypes**: it requires no part annotations for training, and explains its attention naturally – it is looking at *this* region of input because *this* region is similar to *that* prototypical example. Although other attention models focus on similar regions (e.g. head, wing, etc.) as our ProtoPNet, **they cannot be made into a case-based reasoning model like ours**: the only way to find prototypes on other attention models is to analyze *posthoc* what activates a conv-filter of the model most strongly and think of that as a prototype – however, since such prototypes do not participate in the actual model computation, any explanations produced this way are **not always faithful** to the classification decisions.

**Hyperparameter choices (R1, R2, R3):** In our experiments, we chose $H_1 = 1$ and $W_1 = 1$: given that the spatial dimension of conv-output for a $224 \times 224$ image is only $7 \times 7$, a $1 \times 1$ prototype is already large enough to represent a significant part of the original image in the pixel space (we want to learn prototypes **focused on specific parts**). The number of prototypes can be chosen with prior domain knowledge or hyperparameter search: we used 10 prototypes per class, which should be enough to capture a variety of bird parts (or different views of a car). Section S8 of supplement also discusses prototype pruning to remove non-essential prototypes. The result of pruning is a model with *fewer* and *different* number of prototypes for various classes. We also performed an experiment to see the effect of changing the number of prototypes per class: test accuracy of VGG16-based ProtoPNet is $72.4\%$ with 5 prototypes per class, $76.1\%$ with 10 prototypes per class, and $76.2\%$ with 15 prototypes per class. This shows that having too few prototypes limits performance, but having too many prototypes does not further improve accuracy.

**Training algorithm and time complexity (R2):** In the algorithm chart (bottom of page): $w_{\text{base}}$ and $w_{\text{add}}$ denote the parameters of base and additional conv-layers; $N_{\text{SGD}}$ and $N_{\text{convex}}$ denote the number of training epochs in stage 1 and 3; $L$ and $L_{\text{convex}}$ denote the loss function of stage 1 and 3; $\eta_{\text{base}}^{(t)}, \eta_{\text{add}}^{(t)}, \eta_{\text{p}}^{(t)}, \eta_{\text{convex}}$ are learning rates ($t$ denotes epoch #). Prototypes are initialized randomly from uniform distribution over $[0,1]^{H_1 \times W_1 \times D}$ – the last conv-layer uses sigmoid activation, so the conv-features all lie in $[0, 1]$. In our experiments, we set $N_{SGD} = 10$ and $N_{\text{convex}} = 20$. This means that prototype projection happens after every 10 SGD epochs. **Feedforward computation of prototype layer has the same time complexity as that of a regular conv followed by global average pooling**, a configuration common in standard CNNs (e.g. ResNet, DenseNet), because the former takes the max of similarity scores computed over all prototype-sized patches while the latter takes the average of dot-products computed over all filter-sized patches. Similarly, **prototype projection has the same time complexity as feedforward part of SGD on standard conv+pooling**, because the former takes the min distance over all prototype-sized patches, and the latter takes the average of dot-products over all patches. Hence, **using prototype layer (to replace the common conv+pooling in a standard CNN) does not introduce extra time complexity**. For a fixed architecture, time complexity of training/testing ProtoPNet is linear in the number of examples, just like any CNN. Empirically, prototype projection takes $< 250$ seconds for about 6000 training images, **roughly the same time as an SGD epoch** on the same training set using same hardware (1 GPU).

**Other questions: (1) Similarity scores and prediction confidence (R1):** A higher similarity score with a prototype of the predicted class contributes to a more confident final class prediction. **(2) Domains where finding prototypes are useful in itself (R1):** We are currently using this technique to find prototypical tumors in radiology, which can enhance doctors' understanding. **(3) Choice of $L^2$ (R3):** We choose $L^2$ because it is a distance metric that is intuitive to understand, and allows us to easily specify the desired **cluster** and **separation** properties of the latent space.

---

**initialize** $t_0 \leftarrow 0$; $w_{\text{base}} \leftarrow$ weights pre-trained on ImageNet; $w_{\text{add}} \leftarrow$ Kaiming uniform initialization (He et al., 2015);

$\forall j$: prototype $\mathbf{p}_j \leftarrow \text{Uniform}([0,1]^{H_1 \times W_1 \times D})$; $\forall k, j$: $w_h^{(k,j)} \leftarrow 1$ if $\mathbf{p}_j \in \mathbf{P}_k$, $w_h^{(k,j)} \leftarrow 0$ if $\mathbf{p}_j \notin \mathbf{P}_k$;

**while** NOT(converge AND Clst $< -$Sep) **do**

**for** SGD training epoch $t = t_0 + 1, ..., t_0 + N_{\text{SGD}}$ **do**         /* Stage 1:  SGD of layers before the last */

**foreach** batch $[\overline{\mathbf{X}}, \overline{\mathbf{Y}}]$ from $[\mathbf{X}, \mathbf{Y}]$ **do**         /* see Section S9.3 of supplement for learning rates */

$w_{\text{base}} \leftarrow w_{\text{base}} - \eta_{\text{base}}^{(t)} \nabla_{w_{\text{base}}} L(\overline{\mathbf{X}}, \overline{\mathbf{Y}})$; $w_{\text{add}} \leftarrow w_{\text{add}} - \eta_{\text{add}}^{(t)} \nabla_{w_{\text{add}}} L(\overline{\mathbf{X}}, \overline{\mathbf{Y}})$; $\mathbf{P} \leftarrow \mathbf{P} - \eta_{\text{p}}^{(t)} \nabla_{\mathbf{P}} L(\overline{\mathbf{X}}, \overline{\mathbf{Y}})$;

$t_0 \leftarrow t_0 + N_{SGD}$;

**foreach** prototype $\mathbf{p}_j$ **do**         /* Stage 2:  projection of prototypes */

$k \leftarrow$ class of $\mathbf{p}_j$; $\mathbf{p}_j \leftarrow \arg\min_{\mathbf{z} \in \{\tilde{\mathbf{z}}: \tilde{\mathbf{z}} \in \text{patches}(f(\mathbf{x})) \ \forall (\mathbf{x}, y) \in [\mathbf{X}, \mathbf{Y}] \text{ s.t. } y = k\}} \|\mathbf{z} - \mathbf{p}_j\|_2$;

**for** convex training epoch $t' = 1, ..., N_{\text{convex}}$ **do**         /* Stage 3:  convex optimization of last layer */

**foreach** batch $[\overline{\mathbf{X}}, \overline{\mathbf{Y}}]$ from $[\mathbf{X}, \mathbf{Y}]$ **do** $w_h \leftarrow w_h - \eta_{\text{convex}} \nabla_{w_h} L_{\text{convex}}(\overline{\mathbf{X}}, \overline{\mathbf{Y}})$ ;

[Meta-Review · NeurIPS 2019]

The reviewers agreed that this paper includes interesting ideas and is well written. Please read the remaining reviewers' comments carefully for your final version.